# Periodic Materials Generation using Text-Guided Joint Diffusion Model

**Kishalay Das**[1,†]**, Subhojyoti Khastagir**[1]**, Pawan Goyal**[1]**, Seung-Cheol Lee**[2]**,
Satadeep Bhattacharjee**[2]**, Niloy Ganguly**[1]
[1] Indian Institute of Technology, Kharagpur, India
[2] Indo Korea Science and Technology Center, Bangalore, India

## Abstract

Equivariant diffusion models have emerged as the prevailing approach for generating novel crystal materials due to their ability to leverage the physical symmetries of periodic material structures. However, current models do not effectively learn the joint distribution of atom types, fractional coordinates, and lattice structure of the crystal material in a cohesive end-to-end diffusion framework. Also, none of these models work under realistic setups, where users specify the desired characteristics that the generated structures must match. In this work, we introduce TGDMat, a novel text-guided diffusion model designed for 3D periodic material generation. Our approach integrates global structural knowledge through textual descriptions at each denoising step while jointly generating atom coordinates, types, and lattice structure using a periodic-E(3)-equivariant graph neural network (GNN). Extensive experiments using popular datasets on benchmark tasks reveal that TGDMat outperforms existing baseline methods by a good margin. Notably, for the structure prediction task, with just one generated sample, TGDMat outperforms all baseline models, highlighting the importance of text-guided diffusion. Further, in the generation task, TGDMat surpasses all baselines and their text-fusion variants, showcasing the effectiveness of the joint diffusion paradigm. Additionally, incorporating textual knowledge reduces overall training and sampling computational overhead while enhancing generative performance when utilizing real-world textual prompts from experts. Code is available at `https://github.com/kdmsit/TGDMat`

## 1 Introduction

Screening 3D periodic structures and their atomic compositions to identify novel crystal materials with specific chemical properties remains a long-standing challenge in the materials design community. These materials have been fundamental to key innovations such as the development of batteries, solar cells, semiconductors etc. (Butler et al., 2018; Desiraju, 2002). Historically, there have been attempts to generate novel materials by conducting resource-intensive and time-consuming simulations based on Density Functional Theory (DFT) (Kohn & Sham, 1965). Recently, the equivariant diffusion models (Jiao et al., 2023; Luo et al., 2023b; Xie et al., 2021) have demonstrated great potential to generate stable 3D periodic structures of new crystal materials.

However, these models possess several inherent limitations. 1) None of these existing SOTA models learns the joint distribution of atom coordinates, types, and lattice structure of the material through an end-to-end diffusion network. Existing models like CDVAE (Xie et al., 2021) and SyMat (Luo et al., 2023b) learn lattice parameters and atom types separately using a VAE model and further use a score network to learn the conditional distribution of atom coordinates given atom types and lattice. DiffCSP (Jiao et al., 2023), on the other hand, focuses primarily on structure prediction task where it assumes atom types are given and predict the stable crystal structure (lattice and coordinates). 2) Furthermore, these models use SE(3)-equivariant GNNs as backbone denoising network, which largely relies on messages passing around the local neighborhood of the atoms. Hence they fail to incorporate global structural knowledge into the diffusion process, which can enhance the diffusion performance. 3) Finally, these models are unconditional by design. From initial noisy structures without any external constraints, they generate stable crystal structures, which are distributionally

---

†Correspondence to Kishalay: kishalaydas@kgpian.iitkgp.ac.in

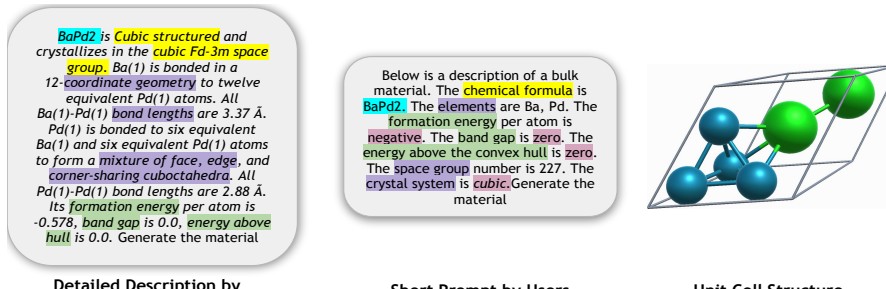

Figure 1: Detailed textual description generated by Robocrystallographer, less-detailed prompts by domain experts, and crystal unit cell structure of $BaPd_2$.

similar to structures of the training dataset. This setup may have limited utility in real-world scenarios, as it lacks a mechanism for users to specify a criteria for the material to be generated. In a realistic setup, users would want to specify certain key details about the target material, like the chemical formula, space group, crystal symmetry, bond lengths, chemical properties, etc as input to the diffusion model, which the generated structure must then match.

In this paper, we propose, TGDMat, a novel *Text-Guided Diffusion Model for Material Generation* that mitigates the limitations mentioned above and enhances the generation capability. Though Text Guided Diffusion Models (TGDMs) produce impressively high-quality data in the form of images (Nichol et al., 2021; Ramesh et al., 2022; Rombach et al., 2022; Saharia et al., 2022), audio (Kreuk et al., 2022; Yang et al., 2022), video (Du et al., 2024), molecules (Gong et al., 2024; Luo et al., 2023a) etc, it remains largely unexplored in periodic material generation. Text-guided diffusion model for new material generation has some key benefits. First, we can leverage popular tools like Robocrystallographer (Ganose & Jain, 2019) to generate a textual description of the material which provides a rich and diverse set of global structural knowledge like chemical formula, lattice constraint, space group number, crystal symmetry, chemical properties, etc. We believe this additional information is helpful for diffusion models in learning underlying crystal geometry. Second, it provides end users the flexibility to use custom prompts to guide the material generation process, ensuring that the resulting material aligns with the user's provided description. Towards that goal, we first develop a diffusion model that jointly generates the atom coordinates, atom types, and lattice structure of crystal materials using a periodic E(3)-equivariant denoising model, satisfying periodic E(3) invariance properties of learned data distribution. Subsequently, we fuse textual information into the reverse diffusion process, which guides the denoising process in predicting material structure as specified by the textual description.

To sum up, our novel contributions in this work are as follows:

- To the best of our knowledge, we are the first to explore text-guided diffusion for material generation. Our proposed TGDMat bridges the gap between natural language understanding and material structure generation.

- Unlike prior models, TGDMat conducts joint diffusion on lattices, atom types, and co-ordinates, enhancing its ability to accurately capture the crystal geometry. Additionally, incorporating global structural knowledge through textual descriptions at each denoising step improves TGDMat's ability to generate plausible materials with valid and stable structures.

- Extensive experiments using popular datasets on benchmark tasks show TGDMat outperforms baseline models with a good margin. Notably, in CSP task, with just one generated sample, TGDMat outperforms all baseline models, highlighting the importance of text-guided diffusion. Moreover, in the generation task, TGDMat outperforms all baselines and their text-fusion variants, showcasing the effectiveness of the joint diffusion paradigm.

- Fusing textual knowledge reduces the overall computational cost for both training and inference of the diffusion model. Moreover, when applied to real-world custom text prompts by experts, TGDMat demonstrates rich generative capability under general textual conditions.

## 2 PRELIMINARIES : CRYSTAL STRUCTURE REPRESENTATION

Crystal material can be modeled by a minimal *unit cell*, which gets repeated infinite times in 3D space on a regular lattice to form the periodic crystal structure. Given a material with $N$ number of atoms in its unit cell, we can describe the unit cell by two matrices: *Atom Type Matrix (A)* and *Coordinate Matrix (X)*. Atom Type Matrix $A = [a_1, a_2, ..., a_N]^T \in \mathbb{R}^{N \times k}$ denotes set of atomic type in one hot representation (k: maximum possible atom types). On the other hand, Coordinate Matrix $X = [x_1, x_2, ..., x_N]^T \in \mathbb{R}^{N \times 3}$ denotes atomic coordinate positions, where $x_i \in \mathbb{R}^3$ corresponds to coordinates of $i^{th}$ atom in the unit cell. Further, there is an additional *Lattice Matrix* $L = [l_1, l_2, l_3]^T \in \mathbb{R}^{3 \times 3}$, which describes how a unit cell repeats itself in the 3D space towards $l_1, l_2$ and $l_3$ direction to form the periodic 3D structure of the material. Formally, a given material can be defined as $M = (A, X, L)$ and we can represent its infinite periodic structure as $\hat{X} = \{\hat{x}_i | \hat{x}_i = x_i + \sum_{j=1}^{3} k_j l_j\}; \hat{A} = \{\hat{a}_i | \hat{a}_i = a_i\}$ where $k_1, k_2, k_3, i \in Z, 1 \le i \le N$.

**Invariances in Crystal Structure.** The basic idea of using generative models for crystal generation is to learn the underlying data distribution of material structure $p(M)$. Since crystal materials satisfy physical symmetry properties (Dresselhaus et al., 2007; Zee, 2016), one of the major challenges here is the learned distribution must satisfy periodic E(3) invariance i.e. invariance to permutation, translation, rotation, and periodic transformations. A formal definition of these invariance properties is provided in Appendix C.

## 3 RELATED WORK: PERIODIC MATERIAL GENERATION

Recently, the majority of the research on material generation focuses on using popular generative models like VAEs (Kingma & Welling, 2013), GANs (Goodfellow et al., 2014) or Diffusion Models (Song & Ermon, 2019; 2020; Ho et al., 2020) to generate 3D periodic structures of materials (Hoffmann et al., 2019; Noh et al., 2019; Ren et al., 2020; Kim et al., 2020; Court et al., 2020; Long et al., 2021; Zhao et al., 2021; Xie et al., 2021; Jiao et al., 2023; Luo et al., 2023b; Zeni et al., 2023; Yang et al., 2023; Jiao et al., 2024; Miller et al., 2024). In specific, state-of-the-art models like CDVAE (Xie et al., 2021) and SyMat (Luo et al., 2023b) combine VAEs and score-based diffusion models to work directly with atomic coordinates, ensuring euclidean and periodic invariance using equivariant graph neural networks(GNNs). Moreover, DiffCSP (Jiao et al., 2023) focuses on structure prediction, jointly optimizing atom coordinates and lattice using a diffusion framework given atomic composition. We provided a comprehensive literature review of other related works in Appendix B.

**Relations with Prior Methods.** Among existing models, DiffCSP comes close to our methodology, however, our work differs in multiple ways. DiffCSP primarily focuses only on the Structure Prediction (*CSP*) task and they didn't explore the Random Generation (*Gen*) task, whereas TGDMat focuses on both tasks. Moreover, unlike DiffCSP, TGDMat can leverage the informative textual descriptions during the denoising process and can jointly learn lattices, atom types, and coordinates, which makes TGDMat a more flexible and robust generative model for new material generation.

## 4 METHODOLOGY

### 4.1 PROBLEM FORMULATION

In this work, given the textual description, we focus on generating a stable crystal structure that aligns with the provided textual description. Formally, given a dataset $\mathcal{M} = \{M_i, T_i\}$, containing crystal structure $M_i = (A_i, X_i, L_i)$ and its text description $(T_i)$, the goal of text guided crystal generation problem is to capture the underlying conditional data distribution $p(M|T)$ via learning a generative model $f_\theta(M|T)$, where $\theta$ is a set of learnable parameters. While training, we need $f_\theta$ to ensure that the learned distribution is invariant to different symmetry transformations mentioned in Section 2. Once trained, given a text description of a plausible material, the learned generative model can sample a valid and stable structure of the material, that is invariant to different symmetry transformations.

### 4.2 TEXTUAL DATASETS

Leveraging textual information to guide the reverse diffusion process remains unexplored in the material design community. To the best of our knowledge, there is currently no text data available

for materials in benchmark databases (mentioned in Section 5.1). Hence, we first curate the textual data of these material databases. Specifically, we propose two approaches for generating textual descriptions of materials, which are easy to follow. First, we utilize a freely available utility tool, *Robocrystallographer* (Ganose & Jain, 2019) to generate detailed textual descriptions about the periodic structure of crystal materials. These descriptions encompass local compositional details like atomic coordination, geometry, etc. as well as global structural aspects like crystal formula, mineral type, space group information, etc. Secondly, we utilized shorter and less detailed prompts that are more easily interpretable by users. We extend the prompt template proposed by (Gruver et al., 2024), which encodes minimal information about the material like its chemical formula, constituent elements, crystal system it belongs to, and its space group number. Further, we specify a few chemical properties, and instead of mentioning their actual values, we provide generic information like negative/positive formation energy, zero/nonzero band gaps, etc. Detailed information regarding the two textual datasets, including their curation process is provided in Appendix D. We have publicly shared the textual datasets for both benchmark material databases with the community for future use.

### 4.3    Proposed Methodology : TGDMat

Our proposed model, TGDMat (Fig. 2), uses an equivariant diffusion model guided by contextual representation of the textual description ($C_p$) to generate a new crystal structure $M = (A, X, L)$. Unlike prior methods (Jiao et al., 2023; Luo et al., 2023b; Xie et al., 2021), our method jointly diffuses $A, X, L$ to learn the underlying data distribution of crystal structure $p(M|C_p)$. Diffusion models (Ho et al., 2020; Song & Ermon, 2019; 2020) are popular generative models that are formulated using a T steps Markov Chain. Given an input crystal material $M_0 = (A_0, X_0, L_0)$, the forward process gradually add noise to $A_0, X_0, L_0$ independently over T steps and the reverse denoising process samples a noisy structure $M_T = (A_T, X_T, L_T)$ from a prior distribution and reconstruct back $M_0$ using some GNN model. At each $t^{th}$ step of denoising ($T \geq t \geq 0$), the contextual representation of the crystal textual description ($C_p$) will guide the diffusion process so that the intermediate structure $M_t$ aligns the target 3D structure constrained on textual conditions. Moreover, the learned distribution of material structure must satisfy periodic E(3) invariance. It is well studied in the literature (Xu et al., 2022) that if the prior distribution $p(x)$ is invariant to a group and the transition probabilities of a Markov chain $y \sim p(y|x)$ exhibit equivariance, the marginal distribution of $y$ at any given time step also remains invariant to group transformations. Hence the learned distribution $p(M_0)$ of the denoising model will satisfy periodic E(3) invariance if the prior distribution $p(M_T)$ is invariant and the neural network used to parameterize the transition probability $q(M_{t-1}|M_t)$ is equivariant to permutational, translation, rotational, and periodic transformations. To satisfy that, we use periodic-E(3)-equivariant GNN model as a backbone denoising network to guide the denoising process. Next in this section, we first explain diffusion on $M$ in 4.3.1, then demonstrate the text-guided denoising network in 4.3.2 and finally training details in 4.4.

#### 4.3.1    Joint Equivariant Diffusion on $M$

**Diffusion on Lattice ($L$).** Since the Lattice Matrix $L = [l_1, l_2, l_3]^T \in \mathbb{R}^{3 \times 3}$ is in continuous space, we leverage the idea of the Denoising Diffusion Probabilistic Model (DDPM) (Ho et al., 2020) for diffusion on $L$. Specifically, given input lattice matrix $L_0 \sim p(L)$, at each $t^{th}$ step, the forward diffusion process iteratively diffuses it through a transition probability $q(L_t|L_0)$ which can be derived as $q(L_t|L_0) = \mathcal{N}(L_t|\sqrt{\bar{\alpha}_t}L_0, (1 - \bar{\alpha}_t)\mathbf{I})$ where, $\bar{\alpha}_t = \prod_{k=1}^{t} \alpha_k$, $\alpha_t = 1 - \beta_t$ and $\{\beta_t \in (0, 1)\}_{t=1}^{T}$ controls the variance of diffusion step following certain noise scheduler. By reparameterization, we can rewrite $L_t = \sqrt{\bar{\alpha}_t}L_0 + \sqrt{1 - \bar{\alpha}_t}\epsilon^L$ where, $\epsilon^L$ is noise sampled from $\mathcal{N}(\mathbf{0}, \mathbf{I})$, added with $L_0$ at $t^{th}$ step to generate $L_t$. After T such diffusion steps, noisy lattice matrix $L_T \sim \mathcal{N}(\mathbf{0}, \mathbf{I})$ is generated. During reverse denoising process, given noisy $L_T \sim \mathcal{N}(\mathbf{0}, \mathbf{I})$ we reconstruct true lattice structure $L_0$ thorough iterative denoising step via learning reverse conditional distribution, which we formulate as $p(L_{t-1}|M_t, C_p) = \mathcal{N}\{L_{t-1}|\mu^L(M_t, C_p), \beta_t \frac{(1-\bar{\alpha}_{t-1})}{(1-\bar{\alpha}_t)}\mathbf{I}\}$ where $\mu^L(M_t, C_p) = \frac{1}{\sqrt{\alpha_t}}(L_t - \frac{1-\alpha_t}{\sqrt{1-\bar{\alpha}_t}}\hat{\epsilon}^L(M_t, C_p, t))$. Intuitively, $\hat{\epsilon}^L$ needs to be subtracted from $L_t$ to generate $L_{t-1}$ and textual representation $C_p$ will steer this reverse diffusion process. We use a text-guided denoising network $\Phi_\theta(A_t, X_t, L_t, t, C_p)$ to model the noise term $\hat{\epsilon}^L(M_t, C_p, t)$. Following the simplified training objective proposed by (Ho et al., 2020), we train denoising model using $l_2$ loss between $\hat{\epsilon}^L$ and $\epsilon^L$

$$\mathcal{L}_{lattice} = \mathbb{E}_{\epsilon^L, t \sim \mathcal{U}(1,T)} \|\epsilon^L - \hat{\epsilon}^L\|_2^2 \tag{1}$$

**Diffusion on Atom Types ($A$).** Prior studies (Jiao et al., 2023; Xie et al., 2021) consider Atom Type Matrix $A$ as the probability distribution for k classes $\in \mathbb{R}^{N \times k}$ (continuous variable) and apply

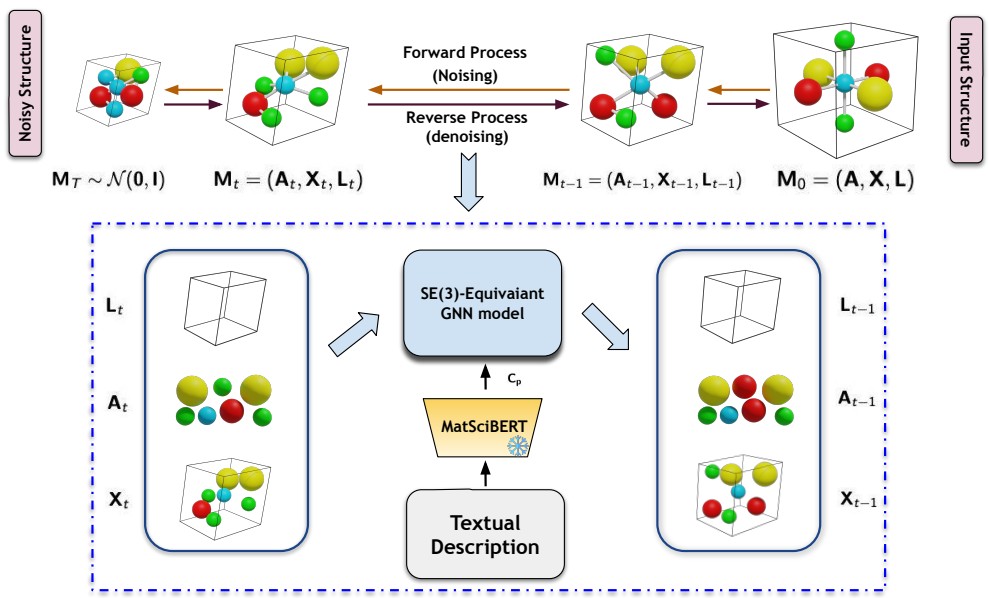

Figure 2: Model Architecture of our proposed text guided diffusion model TGDMat. At $t^{th}$ step of reverse diffusion, given $\boldsymbol{M}_t = (\boldsymbol{A}_t, \boldsymbol{X}_t, \boldsymbol{L}_t)$, we use periodic-E(3)-equivariant GNN model guided by contextual representation of the textual prompts ($\boldsymbol{C_p}$) to generate $\boldsymbol{M}_{t-1} = (\boldsymbol{A}_{t-1}, \boldsymbol{X}_{t-1}, \boldsymbol{L}_{t-1})$

DDPM to learn the distribution. However for discrete data these models are inappropriate and produce suboptimal results (Austin et al., 2021; Campbell et al., 2022). Hence we consider $\boldsymbol{A}$ as N discrete variables belonging to k classes and leverage discrete diffusion model (D3PM) (Austin et al., 2021) for diffusion on $\boldsymbol{A}$. In specific, with $\boldsymbol{a}$ as the one-hot representation of atom $a$, the transition probability for the forward process is $q(\boldsymbol{a}_t|\boldsymbol{a}_{t-1}) = Cat(\boldsymbol{a}_t; \boldsymbol{p} = \boldsymbol{a}_{t-1}\boldsymbol{Q}_t)$, where $Cat(\boldsymbol{a}; \boldsymbol{p})$ is a categorical distribution over $\boldsymbol{a}$ with probabilities $\boldsymbol{p}$ and $\boldsymbol{Q}_t$ is the Markov transition matrix at time step t, defined as $[\boldsymbol{Q}_t]_{i,j} = q(a_t = i|a_{t-1} = j)$. Different choices of $\boldsymbol{Q}_t$ and corresponding stationary distributions are proposed by (Austin et al., 2021) which provides flexibility to control the data corruption and denoising process. We adopted the absorbing state diffusion process, introducing a new absorbing state [MASK] in $\boldsymbol{Q}_t$. At each time step t, an atom either stays in its type state with probability $1 - \beta_t$ or moves to [MASK] state with probability $\beta_t$ and once it moves to [MASK] state, it stays there. Hence, the stationary distribution of this diffusion process has all the mass on the [MASK] state. During denoising process, given textual representation $\boldsymbol{C_p}$, we first sample noisy $\boldsymbol{a}_T$ and obtain $\boldsymbol{a}_0$ thorough iterative denoising step via learning reverse conditional transition $p_\theta(\boldsymbol{a}_{t-1}|\boldsymbol{a}_t, \boldsymbol{C_p}) \propto \sum_{\boldsymbol{a}_0} q(\boldsymbol{a}_{t-1}, \boldsymbol{a}_t|\boldsymbol{a}_0)p_\theta(\boldsymbol{a}_0|\boldsymbol{a}_t, \boldsymbol{C_p})$. We use the text-guided denoising network $\Phi_\theta(\boldsymbol{A}_t, \boldsymbol{X}_t, \boldsymbol{L}_t, t, \boldsymbol{C_p})$ to model this denoising process, which is trained using following loss function :

$$\mathcal{L}_{type} = \mathcal{L}_{VB} + \lambda\mathcal{L}_{CE} \tag{2}$$

where $\mathcal{L}_{VB}$ and $\mathcal{L}_{CE}$ is the variational lower bound and cross-entropy loss respectively and $\lambda$ is a hyperparameter. Details about the diffusion process and the losses $\mathcal{L}_{VB}, \mathcal{L}_{CE}$ are in Appendix E

**Diffusion on Atom Coordinates ($X$).** We can diffuse the Coordinate Matrix $\boldsymbol{X} = [\boldsymbol{x}_1, \boldsymbol{x}_2, ..., \boldsymbol{x}_N]^T \in \mathbb{R}^{N \times 3}$ in two ways: either by diffusing cartesian coordinates or fractional coordinates. Prior works like CDVAE (Xie et al., 2021) and SyMat (Luo et al., 2023b) diffuse cartesian coordinates whereas DiffCSP (Jiao et al., 2023) diffuses fractional coordinates. In our setup, as we are jointly learning atom coordinates and lattice matrix, hence we follow DiffCSP and diffuse fractional coordinates. Fractional coordinates in crystal material resides in quotient space $\mathbb{R}^{N \times 3}/\mathbb{Z}^{N \times 3}$ induced by the crystal periodicity. Since the Gaussian distribution used in DDPM is unable to model the cyclical and bounded domain of $\boldsymbol{X}$, it is not suitable to apply DDPM to model $\boldsymbol{X}$. Hence at each step of forward diffusion, we add noise sampled from Wrapped Normal (WN) distribution (De Bortoli et al., 2022) to $\boldsymbol{X}$ and during denoising leverage Score Matching Networks (Song & Ermon, 2019; 2020)

to model underlying transition probability $q(\boldsymbol{X}_t|\boldsymbol{X}_0) = \mathcal{N}_W(\boldsymbol{X}_t|\boldsymbol{X}_0, \sigma_t^2\mathbf{I})$. In specific, at each $t^{th}$ step of diffusion, we derive $\boldsymbol{X}_t$ as : $\boldsymbol{X}_t = f_w(\boldsymbol{X}_0 + \boldsymbol{\sigma_t}\boldsymbol{\epsilon}^{\boldsymbol{X}})$ where, $\boldsymbol{\epsilon}^{\boldsymbol{X}} \sim \mathcal{N}(\mathbf{0}, \mathbf{I})$, $\boldsymbol{\sigma_t}$ is the noise scheduler and $f_w(.)$ is a truncation function. Given a fractional coordinate matrix $\boldsymbol{X}$, truncation function $f_w(\boldsymbol{X}) = (\boldsymbol{X} - \lfloor \boldsymbol{X} \rfloor)$ returns the fractional part of each element of $\boldsymbol{X}$.

As argued in (Jiao et al., 2023), $q(\boldsymbol{X}_t|\boldsymbol{X}_0)$ is periodic translation equivariant, and approaches uniform distribution $\mathcal{U}(0,1)$ for sufficiently large values of $\boldsymbol{\sigma_T}$. Hence during the denoising process, we first sample $\boldsymbol{X}_T \sim \mathcal{U}(0,1)$ and iteratively denoise via score network for T steps to recover back the true fractional coordinates $\boldsymbol{X}_0$. We use the text-guided denoising network $\Phi_\theta(\boldsymbol{A}_t, \boldsymbol{X}_t, \boldsymbol{L}_t, t, \boldsymbol{C_p})$ to model the denoising process, which is trained using the following score-matching objective function :

$$\mathcal{L}_{coord} = \mathbb{E}_{\substack{\boldsymbol{X}_t \sim q(\boldsymbol{X}_t|\boldsymbol{X}_0) \\ t \sim \mathcal{U}(1,T)}} \|\nabla_{\boldsymbol{X}_t} \log q(\boldsymbol{X}_t|\boldsymbol{X}_0) - \hat{\boldsymbol{\epsilon}}^{\boldsymbol{X}}(\boldsymbol{M}_t, \boldsymbol{C_p}, t)\|_2^2 \tag{3}$$

where $\nabla_{\boldsymbol{X}_t} \log q(\boldsymbol{X}_t|\boldsymbol{X}_0) \propto \sum_{\boldsymbol{K} \in \mathbb{Z}^{N \times 3}} \exp(-\frac{\|\boldsymbol{X}_t - \boldsymbol{X}_0 + \boldsymbol{K}\|_F^2}{2\boldsymbol{\sigma_t}^2})$ is the score function of transitional distribution and $\hat{\boldsymbol{\epsilon}}^{\boldsymbol{X}}(\boldsymbol{M}_t, \boldsymbol{C_p}, t)$ denoising term. More Details are provided in Appendix E

### 4.3.2 TEXT GUIDED DENOISING NETWORK

In this subsection, we will illustrate the detailed architecture of our proposed Text Guided Denoising Network $\Phi_\theta(\boldsymbol{A}_t, \boldsymbol{X}_t, \boldsymbol{L}_t, t, \boldsymbol{C_p})$, which we used during denoising process to generate $\boldsymbol{A}$, $\boldsymbol{X}$ and $\boldsymbol{L}$. As mentioned in 2, the learned distribution of material structure $p(\boldsymbol{M})$ must satisfy periodic E(3) invariance. Hence we leverage a periodic-E(3)-equivariant Graph Neural Network (GNN) integrated with a pre-trained textual encoder to model the denoising process. In particular, as a text encoder, we adopt a pre-trained MatSciBERT (Gupta et al., 2022) model, which is a domain-specific language model for materials science, followed by a projection layer. MatSciBERT is effectively a pre-trained SciBERT model on a scientific text corpus of 3.17B words, which is further trained on a huge text corpus of materials science containing around 285M words. We feed textual description of material $\boldsymbol{T}$ into MatSciBERT and extract embedding of [CLS] token $\boldsymbol{h}_{CLS}$ as a representation of the whole text. Further. we feed $\boldsymbol{h}_{CLS}$ through a projection layer to generate the contextual textual embedding for the material $\boldsymbol{C_p} \in \mathbb{R}^d$, which we pass to the equivariant GNN model to guide the denoising process. Practically, as the backbone network for the denoising process, we extend CSPNet architecture (Jiao et al., 2023), originally developed for crystal structure prediction (CSP) task. CSPNet is built upon EGNN (Satorras et al., 2021), satisfying periodic E(3) invariance condition on periodic crystal structure. At the $k^{th}$ layer message passing, the Equivariant Graph Convolutional Layer (EGCL) takes as input the set of atom embeddings $\boldsymbol{h}^k = [\boldsymbol{h}_1^k, \boldsymbol{h}_2^k, ..., \boldsymbol{h}_N^k]$, atom coordinates $\boldsymbol{x}^k = [\boldsymbol{x}_1^k, \boldsymbol{x}_2^k, ..., \boldsymbol{x}_N^k]$ and Lattice Matrix $\boldsymbol{L}$ and outputs a transformation on $\boldsymbol{h}^{k+1}$. Formally, we can define the $k^{th}$ layer message passing operation as:

$$\boldsymbol{m}_{i,j} = \rho_m\{\boldsymbol{h}_i^k,\ \boldsymbol{h}_j^k,\ \boldsymbol{L}^T\boldsymbol{L},\ \psi_{FT}(\boldsymbol{x}_i^k - \boldsymbol{x}_j^k)\};\ \boldsymbol{m}_i = \sum_{j=1}^N \boldsymbol{m}_{i,j};\ \boldsymbol{h}_i^{k+1} = \boldsymbol{h}_i^k + \rho_h\{\boldsymbol{h}_i^k, \boldsymbol{m}_i\} \tag{4}$$

where $\rho_m, \rho_h$ are MLPs and $\psi_{FT}$ is a Fourier Transformation function applied on relative difference between fractional coordinates $\boldsymbol{x}_i^k, \boldsymbol{x}_j^k$. Fourier Transformation is used since it is invariant to periodic translation and extracts various frequencies of all relative fractional distances that are helpful for crystal structure modeling (Jiao et al., 2023).

We fuse textual representation $\boldsymbol{C_p}$ into input atom feature $\boldsymbol{h}_i^0$ as

$$\boldsymbol{h}_i^0 = \rho \{ f_{atom}(\boldsymbol{a}_i) \mid\mid f_{pos}(t) \mid\mid \boldsymbol{C_p}\} \tag{5}$$

where t is the timestamp of the diffusion model, $f_{pos}(.)$ is sinusoidal positional encoding (Ho et al., 2020; Vaswani et al., 2017), $f_{atom}(.)$ learned atomic embedding function and $\mid\mid$ is concatenation operation. Input atom features $\boldsymbol{h}^0$ and coordinates $\boldsymbol{x}^0$ are fed through $\mathcal{K}$ layers of EGCL to produce $\hat{\boldsymbol{\epsilon}}^{\boldsymbol{L}}$, $p(\boldsymbol{A}_{t-1}|\boldsymbol{M}_t)$ and $\hat{\boldsymbol{\epsilon}}^{\boldsymbol{X}}$ as follows :

$$\hat{\boldsymbol{\epsilon}}^{\boldsymbol{L}} = \boldsymbol{L}\rho_L(\frac{1}{N}\sum_{i=1}^N \boldsymbol{h}^{\mathcal{K}});\ p(\boldsymbol{A}_{t-1} \mid \boldsymbol{M}_t) = \rho_A(\boldsymbol{h}^{\mathcal{K}});\ \hat{\boldsymbol{\epsilon}}^{\boldsymbol{X}} = \rho_X(\boldsymbol{h}^{\mathcal{K}}) \tag{6}$$

where $\rho_L, \rho_A, \rho_X$ are MLPs on the final layer embeddings. Intuitively, we feed global structural knowledge about the crystal structure into the network by injecting contextual representation $\boldsymbol{C_p}$ into input atom features. This added signal participates through message-passing operations in Eq. 4 and guides in denoising atom types, coordinates, and lattice parameters such that it can capture the global crystal geometry and aligned with the input stable structure specified by textual description.

| Method | # Samples | Perov-5 | | Carbon-24 | | MP-20 | |
|---|---|---|---|---|---|---|---|
| | | Match Rate ↑ | RMSE ↓ | Match Rate ↑ | RMSE ↓ | Match Rate ↑ | RMSE ↓ |
| P-cG-SchNet | 1 | 48.22 | 0.4179 | 17.29 | 0.3846 | 15.39 | 0.3762 |
| | 20 | 97.94 | 0.3463 | 55.91 | 0.3551 | 32.64 | 0.3018 |
| CDVAE | 1 | 45.31 | 0.1138 | 17.09 | 0.2969 | 33.90 | 0.1045 |
| | 20 | 88.51 | 0.0464 | 88.37 | 0.2286 | 66.95 | 0.1026 |
| DiffCSP | 1 | 52.02 | 0.0760 | 17.54 | 0.2759 | 51.49 | 0.0631 |
| | 20 | **98.60** | 0.0128 | 88.47 | 0.2192 | 77.93 | 0.0492 |
| TGDMat (Short) | 1 | 56.54 | 0.0583 | 24.13 | 0.2424 | 52.22 | 0.0597 |
| | 20 | 98.25 | 0.0137 | 88.28 | 0.2252 | 80.97 | 0.0443 |
| TGDMat (Long) | 1 | **90.46** | **0.0203** | **44.63** | **0.2266** | **55.15** | **0.0572** |
| | 20 | 98.59 | **0.0072** | **95.27** | **0.1534** | **82.02** | **0.0483** |

Table 1: Summary of results on *CSP* task. We highlight the best and second-best performances in bold and underlined, respectively.

## 4.4 TRAINING AND SAMPLING

TGDMat is trained using the following combined loss:
$$\mathcal{L} = \lambda_L \mathcal{L}_{lattice} + \lambda_A \mathcal{L}_{type} + \lambda_X \mathcal{L}_{coord} \tag{7}$$
where $\mathcal{L}_{lattice}$, $\mathcal{L}_{type}$ and $\mathcal{L}_{coord}$ are lattice $l_2$ loss (Eq. 1), type loss (Eq. 2) and coordinate score matching loss (Eq. 3) respectively and $\lambda_L, \lambda_A, \lambda_X$ are hyperparameters control the relative weightage between these different loss components. During training, we freeze the MatSciBERT parameters and do not tune them further. During sampling, we use the Predictor-Corrector (Song et al., 2020) sampling mechanism to sample $A_0$, $X_0$ and $L_0$. Training/Sampling algorithms are provided in Appendix E.5

## 5 EXPERIMENTS

We provide a comprehensive evaluation of our method against several baselines on two benchmark tasks. First, in 5.1, we provide a brief overview of the experimental setup, including benchmark tasks, and datasets. Next, in 5.2, we demonstrate how textual data enhances the prediction of stable crystal structures. Following that, in 5.3, we highlight the effectiveness of our proposed joint diffusion paradigm in enhancing its ability to generate novel crystal materials. Additionally, we present the correctness of our generated materials 5.4, the computational cost of training and inference 5.5, and a few ablation studies 5.6. Appendix F describes additional experiments and more ablations studies.

## 5.1 BENCHMARK TASKS AND DATASETS

Following the prior works (Xie et al., 2021; Jiao et al., 2023), we evaluate our proposed model TGDMat on two benchmark tasks for material generation, *Crystal Structure Prediction (CSP)* and *Random Material Generation (Gen)*, using three popular material datasets: **Perov-5** (Castelli et al., 2012a;b), **Carbon-24** (Pickard., 2020) and **MP-20** (Jain et al., 2013b). We curated textual data for these datasets with a textual description of each material. Specifically, we generate both long detailed textual descriptions and shorter prompts using approaches mentioned in 4.2. While training TGDMat, we split the datasets into the train, test, and validation sets following the convention of 60:20:20 (Xie et al., 2021). More details about the dataset and the experimental setup are in Appendix F.1

## 5.2 CRYSTAL STRUCTURE PREDICTION (CSP)

**Setup.** In *CSP* task, the goal is to predict the crystal structure (atom coordinates and lattice) given atom types. In text guidance setup, TGDMat utilizes textual descriptions during the denoising process and jointly predicts atom coordinates and lattice parameters from randomly sampled noise. To assess TGDMat's effectiveness, we choose three SOTA generative models: **P-cG-SchNet** (Gebauer et al., 2022), **CDVAE** (Xie et al., 2021), and **DiffCSP** (Jiao et al., 2023). Following prior works (Jiao et al., 2023; Xie et al., 2021), we evaluate the performance of all competing models using standard metrics **Match Rate (MR)** and **RMSE**, by matching the generated structure and the ground truth structure in the test set. In particular, we generate k samples for each material structure in the test set, and determine the matching metrics (MR and RMSE) if at least one sample aligns with the ground truth structure. Details about evaluation metrics in Appendix F.2

| Dataset | Method | Validity ↑ | | Coverage ↑ | | Property Statistics (EMD) ↓ | | |
| --- | --- | --- | --- | --- | --- | --- | --- | --- |
| | | Compositional | Structural | COV-R | COV-P | # Element | $\rho$ | $\mathcal{E}$ |
| Perov-5 | CDVAE | 98.59 | **100** | 99.45 | 98.46 | 0.0628 | 0.1258 | 0.0264 |
| | CDVAE+ | 98.45 | 99.8 | 99.53 | 99.09 | 0.0609 | 0.1276 | 0.0223 |
| | SyMat | 97.40 | **100** | 99.68 | 98.64 | 0.0177 | 0.1893 | 0.2364 |
| | SyMat+ | 97.88 | 99.9 | 99.70 | 98.79 | 0.0172 | 0.1755 | 0.2566 |
| | DiffCSP | **98.85** | **100** | 99.74 | 98.27 | 0.0128 | 0.1110 | 0.0263 |
| | DiffCSP+ | 98.44 | **100** | 99.85 | 98.53 | 0.0119 | 0.1070 | 0.0241 |
| | TGDMat(Short) | 98.28 | **100** | 99.7 | 99.24 | 0.0108 | 0.0947 | 0.0257 |
| | TGDMat(Long) | 98.63 | **100** | **99.83** | **99.52** | **0.0090** | **0.0497** | **0.0187** |
| Carbon-24 | CDVAE | - | **100** | 99.8 | 83.08 | - | 0.1407 | 0.285 |
| | CDVAE+ | - | **100** | 99.8 | 84.76 | - | 0.1377 | 0.266 |
| | SyMat | - | **100** | **99.9** | 97.59 | - | 0.1195 | 3.9576 |
| | SyMat+ | - | **100** | **99.9** | **97.63** | - | 0.1171 | 3.862 |
| | DiffCSP | - | **100** | **99.9** | 97.27 | - | 0.0805 | 0.082 |
| | DiffCSP+ | - | **100** | **99.9** | 97.33 | - | 0.0763 | 0.085 |
| | TGDMat(Short) | - | **100** | 99.8 | 91.77 | - | 0.0681 | 0.087 |
| | TGDMat(Long) | - | **100** | **99.9** | 92.43 | - | **0.043** | **0.063** |
| MP-20 | CDVAE | 86.70 | **100** | 99.15 | 99.49 | 1.432 | 0.6875 | 0.2778 |
| | CDVAE+ | 87.42 | **100** | 99.57 | 99.81 | 0.972 | 0.6388 | 0.2977 |
| | SyMat | 88.26 | **100** | 98.97 | **99.97** | 0.5067 | 0.3805 | 0.3506 |
| | SyMat+ | 88.47 | 99.9 | 99.01 | 99.95 | 0.4865 | 0.3879 | 0.3489 |
| | DiffCSP | 83.25 | **100** | 99.71 | 99.76 | 0.3398 | 0.3502 | 0.1247 |
| | DiffCSP+ | 85.07 | **100** | 99.8 | 99.89 | 0.3122 | 0.3799 | 0.1355 |
| | TGDMat(Short) | 86.60 | **100** | 99.79 | 99.88 | 0.3337 | **0.3296** | **0.1154** |
| | TGDMat(Long) | **92.97** | **100** | **99.89** | 99.95 | **0.2890** | 0.3382 | 0.1189 |

Table 2: Summary of results on *Gen* task, with the best and second-best performances in bold and underlined, respectively. The table contains "-" values for metrics that don't apply to certain datasets.

**Results and Discussions.** We report the Match Rate and RMSE of all the baselines and TGDMat for three benchmark datasets in Table 1. We trained TGDMat using both detailed textual descriptions and short prompts, as outlined in 4.2 and report them as **TGDMat(Long)** and **TGDMat(Short)** respectively in Table 1. We observe that both variants of TGDMat surpass all the baseline models with a good margin across three datasets, which shows the rich capability of text-guided joint diffusion to predict stable crystal structure. Notably, while prior diffusion models demonstrate improved Match Rates and lower RMSE when generating 20 samples per test material, they largely fail in both metrics when generating only one sample per test material. However, generating 20 samples per test material to match the structures is unrealistic and computationally burdensome. This highlights the importance of text-guided diffusion: incorporating textual knowledge during reverse diffusion aids in aligning the noisy structure with the 3D geometry of stable realistic materials. Specifically, with just one generated sample ($k = 1$) per test material, both the variants of TGDMat outperform all baseline models, thereby reducing computational overhead. Moreover, even with 20 generated samples ($k = 20$) performance of TGDMat is significantly better for Carbon-24 and MP-20, whereas comparable with DiffCSP for Perov-5. In general, due to the extensive metadata provided by the detailed description about the crystal structure, the performance enhancement in the long variant surpasses that of the short one. However, these findings collectively demonstrate the effectiveness of text-guided diffusion in the stable crystal structure prediction task.

## 5.3 RANDOM MATERIAL GENERATION (GEN)

**Setup.** In *Gen* task, the goal is to generate novel stable materials (both structure and atom types), that are distributionally similar to the materials in the test dataset. In TGDMat, by design choice, we use the textual description of crystal materials in the test dataset during the reverse diffusion process to enhance the generation capability. To evaluate performance of TGDMat in this task, we choose three popular state-of-the-art generative models: **CDVAE** (Xie et al., 2021), **SyMat** (Luo et al., 2023b), and **DiffCSP** (Jiao et al., 2023). For a fair comparison, we also consider their text-guided variants such as **CDVAE+**, **SyMat+** and **DiffCSP+** respectively, where we fuse the contextual representation of (Long) text data into those models using the same process described in 4.3.2. Following (Xie et al., 2021), for evaluating all the competing models, we use seven metrics under three broad categories: **Validity**, **Coverage**, and **Property Statistics** (Details in Appendix F.2). To ensure a fair comparison regarding sample size, we generate a number of samples equal to the test data size for all baseline models, both unconditional and text-guided variants, and evaluate their performance accordingly.

**Results and Discussions.** We report the result of **TGDMat (Long and Short)** and all the

| Long (Detailed) Description | Short Prompt | Ground truth | Generated Samples |
|---|---|---|---|
| LaNi2Ge2 crystallizes in the tetragonal I4/mmm space group. La is bonded in a 16-coordinate geometry to eight equivalent Ni and eight equivalent Ge atoms. All La-Ni bond lengths are 3.25 Å. ... The Ge-Ge bond length is 2.66 Å. The formation energy is -0.691. The band gap is 0.0. The energy above the convex hull is 0.0. | Below is a description of a bulk material. The chemical formula is La(NiGe)2. The elements are La, Ni, Ge. The formation energy is negative. The band gap is zero. The energy above the convex hull is zero. The spacegroup number is 138. The crystal system is tetragonal. Generate the material. | | |
| HgScNOF is alpha Rhenium trioxide-derived structured and crystallizes in the orthorhombic Pmmm space group. The structure consists of one Hg cluster inside a ScNOF ... linear geometry to two equivalent Sc atoms. The formation energy is 1.1428. | Below is a description of a bulk material. The chemical formula is HgScNOF. The elements are Sc,Hg,N,O,F. The formation energy is positive. The spacegroup number is 46. The crystal system is orthorhombic. Generate the material. | | |

Table 3: Visualization of the generated structures given textual description. Note that our model produces rotated or translated versions of the ground truth material, owing to the periodic-E(3)invariance.

baseline models in Table 2. We observe that both variants of TGDMat consistently enhances performance across almost all metrics across the benchmark datasets. Particularly on the Perov-5 dataset, TGDMat outperforms all baseline models across all metrics except for compositional validity, where its performance is on par with state-of-the-art results. In the Carbon-24 and MP-20 datasets, TGDMat exhibits performance improvements across all metrics except for COV-P. Additionally, our experiments indicate that utilizing shorter prompts results in a slight decrease in overall performance compared to the longer variant. Nonetheless, the performance remains comparable to baseline models. Overall, TGDMat exhibits promising performance in the Random Material Generation task, indicating its effectiveness in integrating global textual knowledge into the reverse diffusion process to generate more stable periodic structures of 3D crystal materials. Moreover TGDMat outperforms text guided variant of baseline models CDVAE+, SyMat+ and DiffCSP+ as well, which highlights the effectiveness of using joint diffusion to learn $A$,$X$ and $L$.

Additionally, we present visualizations of a few generated materials based on the textual descriptions in Table 3 and compare them with the ground truth material structure. The generated samples exhibit clear matches to the ground truth structure, highlighting the generation capability of the TGDMat model given information in text form. More visualization results are in Appendix F.7.

## 5.4 Correctness of Generated Materials

**Setup.** In this section, we investigate whether the generated material matches different features specified by the textual prompts. TGDMat has the capability to process textual prompts given by the user, enabling it to manage global attributes about crystal materials such as Formula, Space group, Crystal System, and different property values like formation energy, band-gap, etc. To ensure the fidelity of our model's outputs concerning these specified global attributes from the text prompt, We randomly generated 1000 materials (sampled from all three Datasets) based on their respective textual descriptions (both Long and Short) and assessed the percentage of generated materials that matched the global features outlined in the text prompt. In specific, we matched the Formula, Space group, and Crystal System of generated materials with the textual descriptions. Moreover, we examined whether properties such as formation energy and bandgap matched the specified criteria as per the text prompt (positive/negative, zero/nonzero).

**Results and Discussions.** We report the results for TGDMat using long prompt in Table 4 and short prompts in Table 8 (F.4). In general, using longer text, considering Perov-5 and Carbon-24 datasets, the generated material meets the specified criteria effectively. However, when dealing with the MP-20 dataset, which is more intricate due to its complex structure and composition, performance tends to decline. Additionally, when using shorter prompts, overall performance suffers across all datasets compared to longer text inputs. This is because the longer text, provided by the robocrystallographer, offers a comprehensive range of information, both global and local

## 5.5 Computational Cost for Training and Sampling

Integrating textual knowledge during reverse diffusion for crystal material generation offers a key advantage: it accelerates convergence towards realistic structures and reduces computational overhead. We observe, compared to other baseline models, TGDMat incurs substantially lower computation costs during training and sampling processes. Compared to baseline models, our approach notably

| | Global Features in Text Prompt | % of Matched Materials | | |
|---|---|---|---|---|
| | | Perov-5 | Carbon-24 | MP-20 |
| TGDMat(Long) | Formula | 97.50 | 98.20 | 70.54 |
| | Space Group | 87.00 | 80.79 | 67.88 |
| | Crystal System | 92.60 | 91.55 | 73.54 |
| | Formation Energy | 95.49 | - | 92.88 |
| | Band Gap | - | 98.61 | 96.73 |

Table 4: Correctness of generated materials matching conditions specified by the textual prompts.

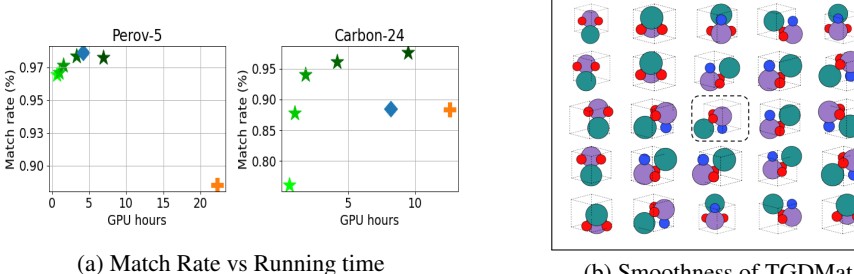

(a) Match Rate vs Running time

(b) Smoothness of TGDMat

Figure 3: (a) Match Rate vs Running time (GPU Hours) for different variants of TGDMat(Long) {50 Steps ★, 100 Steps ★, 200 Steps ★, 500 Steps ★, 1K Steps ★ }, DiffCSP ◆ and CDVAE ✚. (b) Materials sampled given the textual description of the center ground truth material $\boxed{\mathbf{M}}$. The sampled materials are structurally similar (rotated or translated) to each other as well as the ground truth.

cuts down on training time, requiring only 500 epochs compared to 3K or 4K epochs for CDVAE and DiffCSP on Perov-5 and Carbon-24 datasets respectively. Additionally, our method reduces sampling steps, making it faster to generate new structures. While CDVAE and DiffCSP need 5K and 1K steps respectively, our model only requires 500 steps. We compare the performance of CDVAE and DiffCSP with different TGDMat(Long) variants with 50, 100, 200, 500, and 1K steps and report the match rate of the predicted crystal structure vs running time (GPU hours in P100 GPU server) for Perov-5 and Carbon-24 datasets in Fig. 3(a). We notice that the inference time for CDVAE is lengthier as it necessitates 5K steps for each generation. However, for Carbon-24, TGDMat with 200 or 500 steps outperforms DiffCSP with 1K steps. Additionally, for Perov-5, TGDMat with 500 steps achieves results comparable to DiffCSP with 1K steps.

## 5.6 SMOOTHNESS OF MODEL'S GENERATION

We qualitatively demonstrate the smoothness of the crystal generation process in our model. We provide a textual description of a ground truth material and generate several samples of materials to assess the diversity of the generated structures. Figure 3(b) summarizes the results for one crystal material from the Perov-5 dataset, which shows that the generated materials are structurally similar to each other and the given input material.

## 6 CONCLUSION

In this work, we explore a practical approach of generating stable crystal materials given a textual description of the material. We propose TGDMat, which jointly diffuse atom types, fractional coordinates, and lattice structure for crystal materials using a periodic-E(3)-equivariant denoising model. We further integrate textual information into the reverse diffusion process through a pre-trained transformer model, which guides the denoising process in learning the crystal 3D geometry matching the specification by textual description. Extensive experiments conducted on two benchmark generative tasks reveal that TGDMat surpasses all popular baseline models by a good margin. Furthermore, integrating textual knowledge reduces the overall computational cost for both training and inference of the diffusion model. Moreover, when applied to real-world custom text prompts by experts, TGDMat demonstrates rich generative capability under general textual conditions.

## 7 ACKNOWLEDGMENT

This work was funded by Indo Korea Science and Technology Center, Bangalore, India, under the project name "Generating Stable Periodic Materials using Diffusion Models". We thank the Ministry of Education, Govt of India, for supporting Kishalay with Prime Minister Research Fellowship (PMRF) during his Ph.D. tenure.

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

PERIODIC MATERIALS GENERATION USING TEXT-GUIDED JOINT DIFFUSION MODEL (TECHNICAL APPENDIX)

## A  LIMITATIONS AND FUTURE WORK

1) One of the major limitations and scope of the future work of our proposed work is the lack of independent textual datasets for material generation tasks. In our experimental setup, our model relied on textual data extracted from existing datasets. Initially, we extracted text data from CIF files of materials in the test sets of Perov-5, Carbon-24, and MP-20, utilizing this data to evaluate our model and all baseline models. While the experimental results show promise, a more robust evaluation could have been achieved with an independent dataset containing only textual prompts. This would enable us to assess how effectively these models can generate the underlying 3D structure of materials through a text-guided diffusion process. Hence curating an independent textual dataset for material generation containing a diverse set of meta-information will be a future scope for research.

2) Given text prompts/descriptions, we generate contextual representation using a text encoder in TGDMat, where we adopted a pre-trained MatSciBERT (Gupta et al., 2022) model, which is a domain-specific language model for materials science. Also, while training TGDMat, we freeze the MatSciBERT parameters and do not tune them further. Moreover, during sampling, the user must follow a specific format (Long/Short) to provide the text description of the target material. This setup limits the expressive power of the textual representation. We investigated the robustness of TGDMat with much shorter prompts to sample from pre-trained TGDMat model in F.5, but observed performance degradation across all the benchmark dataset on *Gen* task. Hence, Exploring state-of-the-art LLMs and further fine-tuning them during training may create more powerful text conditional diffusion models and provide flexibility to process text prompts of different formats. However, that might create computational overhead as it will increase the number of parameters significantly. This provides scope for further investigation and we keep it as scope for future work.

## B  MORE RELATED WORK

### B.1  CRYSTAL REPRESENTATION LEARNING

In recent times, graph neural network (GNN) based approaches have emerged as a powerful model in learning robust representation of crystal materials, which enhance fast and accurate property prediction. CGCNN (Xie & Grossman, 2018) is the first proposed model, which represents a 3D crystal structure as an undirected weighted multi-edge graph and builds a graph convolution neural network directly on the graph. Following CGCNN, there are a lot of subsequent studies (Chen et al., 2019; Choudhary & DeCost, 2021; Das et al., 2023a; Louis et al., 2020; Park & Wolverton, 2020; Schmidt et al., 2021), where authors proposed different variants of GNN architectures for effective crystal representation learning. Recently, graph transformer-based architecture Matformer (Yan et al., 2022) is proposed to learn the periodic graph representation of the material, which marginally improves the performance, however, is much faster than the prior SOTA model. Moreover, scarcity of labeled data makes these models difficult to train for all the properties, and recently, some key studies (Das et al., 2022; 2023b) have shown promising results to mitigate this issue using transfer learning, pre-training, and knowledge distillation respectively.

### B.2  DIFFUSION MODELS

The fundamental idea of the diffusion model, as initially proposed by (Sohl-Dickstein et al., 2015), is to gradually corrupt data with diffusion noise and learn a neural model to recover back data from noise. Idea of diffusion further developed in two broad categories - 1) *Score Matching Network* (Song & Ermon, 2019; 2020) and 2) *Denoising Diffusion Probabilistic Models (DDPM)* (Ho et al., 2020). In recent times diffusion models have emerged as a powerful new family of deep generative models, achieving remarkable performance records across numerous applications such as image synthesis (Dhariwal & Nichol, 2021; Ramesh et al., 2022; Rombach et al., 2022), molecular conformer generation (Shi et al., 2021; Xu et al., 2022), molecular graph generation (Liu et al., 2021), protein folding (Luo et al., 2022; Wu et al., 2021) etc.

### B.3  CONDITIONAL DIFFUSION MODELS

The initial DDPM model (Ho et al., 2020) demonstrated unconditional diffusion models for image generation, where the output cannot be directed towards a desired characteristic or property. In guided

|  | DiffCSP | TGDMat |
|---|---|---|
| Tasks | Only CSP Task | Both CSP and Gen Tasks |
| Diffusion on Atom Type | - | Discrete Diffusion (D3PM) |
| Model Category | Unconditional; unable to specify the criteria required by the user | Conditional; able to specify the criteria required by the user (in Text Format) |
| Text Guided Diffusion | No | Yes |

Table 5: Key Differences between TGDMat from DiffCSP

diffusion models, the sampling process can be steered by a prompt, which can be a textual description of the desired output, reference image, or any other type of media.

In the field of image generation by diffusion models, Ramesh et al (Ramesh et al., 2022) came up with a text-guided diffusion model called Dall-E2 which showed how textual prompt can be used to steer the sampling process. While training the model, both the image and its textual description are encoded and mapped together, and the encoding of the prompt is used to generate the image during sampling. Another way of guiding the diffusion process using a separate classifier model was shown by (Dhariwal & Nichol, 2021). They trained a classifier on the noised images and used the gradient of the classifier to guide the sampling process. In the classifier-free setting, (Ho & Salimans, 2022) trained two diffusion models, one guided and one unguided, and combined the resulting score estimated during sampling to get the desired outcome. OpenAI's CLIP (Radford et al., 2021) further improved the relevance of the generated image to the given prompt by scoring the correctness of the generated image given the textual prompt.

Similar efforts have been made in the field of molecular generative models. The shortcoming of SMILES-based autoregressive models were addressed by TGM-DLM (Gong et al., 2024) by utilizing diffusion models. This necessitates a two step process, text-guided generation phase, where the SMILES representation is generated from Gaussian noise with the help of a textual description, and correction phase, where necessary rectification are made for the correctness of SMILES string format. This is one of the drawbacks of the SMILES string format, which was addressed by 3M-Diffusion (Zhu et al., 2024), where they have generated molecular graphs from a given textual description.

### B.4 CRYSTAL MATERIAL GENERATION

In the past, there were limited efforts in creating novel periodic materials, with researchers concentrating on generating the atomic composition of periodic materials while largely neglecting the 3D structure. With the advancement of generative models, the majority of the research focuses on using popular generative models like VAEs or GANs to generate 3D periodic structures of materials, however, they either represent materials as three-dimensional voxel images (Court et al., 2020; Hoffmann et al., 2019; Long et al., 2021; Noh et al., 2019) and generate images to depict material structures (atom types, coordinates, and lattices), or they directly encode material structures as embedding vectors (Kim et al., 2020; Ren et al., 2020; Zhao et al., 2021). However, these models neither incorporate stability in the generated structure nor are invariant to any Euclidean and periodic transformations. In recent times equivariant diffusion models (Xie et al., 2021; Luo et al., 2023b; Jiao et al., 2023; Yang et al., 2023; Jiao et al., 2024; Miller et al., 2024) have become the leading method for generating stable crystal materials, thanks to their capability to utilize the physical symmetries of periodic material structures. In specific, state-of-the-art models like CDVAE (Xie et al., 2021) and SyMat (Luo et al., 2023b) integrate a variational autoencoder (VAE) and powerful score-based decder network, work directly with the atomic coordinates of the structures and uses an equivariant graph neural network to ensure euclidean and periodic invariance. However, both CDVAE and SyMat first predict the lattice parameters and atomic composition using the VAE model and subsequently update the coordinates using score based diffusion model. Moreover, given atomic composition, DiffCSP (Jiao et al., 2023) jointly optimizes the atom coordinates and lattice using a diffusion framework to predict the crystal structure with high precision.

**Relations with Prior Methods.** Among the existing models, DiffCSP (Jiao et al., 2023) comes close to our methodology, however our work differs from it in multiple ways. DiffCSP primarily focuses only on the Crystal Structure Prediction (CSP) task and they didn't explore

the Crystal Generation task, whereas TGDMat focuses on both tasks. Moreover, unlike DiffCSP, TGDMat can leverage the informative textual descriptions during the reverse diffusion process and can jointly learn lattices, atom types, and fractional coordinates from randomly sampled noise. This makes TGDMat more flexible and robust in Crystal Generation and Structure Prediction tasks.

## C  INVARIANCES IN CRYSTAL STRUCTURE

The basic idea of using generative models for crystal generation is to learn the underlying data distribution of material structure $p(\boldsymbol{M})$. Since crystal materials satisfy physical symmetry properties (Dresselhaus et al., 2007; Zee, 2016), one of the major challenges here is the learned distribution must satisfy periodic E(3) invariance i.e. invariance to permutation, translation, rotation, and periodic transformations.

- ***Permutation Invariance :*** If we permute the indices of constituent atoms it will not change the material. Formally, given any material $\boldsymbol{M} = (\boldsymbol{A}, \boldsymbol{X}, \boldsymbol{L})$, using any permutation matrix $\mathbf{P}$ if we permute $\boldsymbol{A}$ and $\boldsymbol{X}$ as $\mathbf{P}(\boldsymbol{A})$ and $\mathbf{P}(\boldsymbol{X})$, then new material $\boldsymbol{M_P} = (\mathbf{P}(\boldsymbol{A}), \mathbf{P}(\boldsymbol{X}), \boldsymbol{L})$ will remains unchanged. Hence the underlying distribution is also the same i.e $p(\boldsymbol{M}) = p(\boldsymbol{M_P})$.

- ***Translation Invariance :*** If we translate the atom coordinates by a random vector it will not change the structure of the material. Formally, given any material $\boldsymbol{M} = (\boldsymbol{A}, \boldsymbol{X}, \boldsymbol{L})$, if we translate $\boldsymbol{X}$ by an arbitrary translation vector $\mathbf{u} \in \mathbb{R}^3$, new generated material $\boldsymbol{M_P} = (\boldsymbol{A}, \boldsymbol{X} + \mathbf{u}\mathbf{1}^T, \boldsymbol{L})$ will be the same as $\boldsymbol{M}$. Hence $p(\boldsymbol{M}) = p(\boldsymbol{M_T})$ must satisfy.

- ***Rotational Invariance :*** If we rotate the atom coordinates and lattice matrix, the material remains unchanged. Formally, using any orthogonal rotational matrix $\mathbf{Q} \in R^{3 \times 3}$ (satisfying $\mathbf{Q}^T \mathbf{Q} = \mathbf{I}$), if we rotate $\boldsymbol{X}$ and $\boldsymbol{L}$ of any material $\boldsymbol{M}$ and generate new $\boldsymbol{M_R} = (\boldsymbol{A}, \boldsymbol{QX}, \boldsymbol{QL})$, then actually different representations of the same material. Hence $p(\boldsymbol{M}) = p(\boldsymbol{M_R})$ must satisfy.

- ***Periodic Invariance :*** Finally, since the atoms in the unit cell can periodically repeat itself infinite times along the lattice vector, there can be many choices of unit cells and coordinate matrices representing the same material. Formally, given coordinates $\boldsymbol{X}$, after applying periodic transformation using random matrix $\boldsymbol{K} \in R^{n \times 3}$, new coordinates $\mathbf{X}' = \boldsymbol{X} + \boldsymbol{KL}$ are periodically equivalent. Hence $\boldsymbol{M} = (\boldsymbol{A}, \boldsymbol{X}, \boldsymbol{L})$ and $\mathbf{M'} = (\boldsymbol{A}, \mathbf{X}', \boldsymbol{L})$ are same material and $p(\boldsymbol{M}) = p(\mathbf{M'})$ must hold.

## D  TEXTUAL DATASET

Leveraging textual information to guide the reverse diffusion process remains unexplored in the material design community. To the best of our knowledge, there is currently no dataset available that includes textual descriptions of the materials present in standard benchmark databases (Section 5.1) used for material generation. In specific, we propose two methods for generating textual descriptions of materials. Hence, we first curate the textual dataset containing textual descriptions of these materials to train our model.

***Long Detailed Textual Description:***  First, we utilize a freely available utility tool known as *Robocrystallographer* (Ganose & Jain, 2019) to generate detailed textual descriptions about the periodic structure of crystal materials encoded in Crystallographic Information Files (CIF Files). Robocrystallographer breaks down crystal structures into two main components: local compositional details such as atomic coordination, geometry, polyhedral connectivity, and tilt angles, as well as global structural aspects like crystal formula, mineral type, space group information, symmetry, and dimensionality. This information is presented in three formats: JSON for machine processing, human-readable text for easy comprehension akin to descriptions provided by humans, and machine learning format for specialized analysis. We choose the human-readable text format to compile textual datasets, which closely resemble descriptions given of the crystal structure by humans.

***Short Custom Prompts:***  Secondly, we utilized shorter and less detailed prompts that are more easily interpretable by users. We extend the prompt template proposed by (Gruver et al., 2024), which encodes minimal information about the material like its chemical formula, constituent elements, crystal system it belongs to, and its space group number. Further, we specify a few chemical properties, and instead of mentioning their actual values, we provide generic information

like negative/positive formation energy, zero/nonzero band gaps, etc. We used the Pymatgen tool (Ong et al., 2013) to extract this information from the Crystallographic Information Files (CIF Files) and curate the textual prompts.

An illustrative example of both these textual descriptions and the unit cell structure is provided in Figure 4.

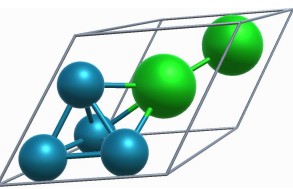

**Detailed Description by Robocrystallographer**     **Short Prompt by Users**     **Unit Cell Structure**

Figure 4: Detailed textual description generated by Robocrystallographer, short/less-detailed prompts by experts, and crystal unit cell structure of $\mathbf{BaPd_2}$ from Material Projects dataset. Text generated by Robocrystallographer contains both local chemical compositional information related to atom/bonds (like site coordination, geometry, polyhedral connectivity, and tilt angles) and global structural knowledge (like mineral type, space group information, symmetry, and dimensionality).The shorter prompt encodes minimal information about the material like its chemical formula, constituent elements, crystal system, and few chemical properties.

# E  JOINT EQUIVARIANT DIFFUSION ON $\boldsymbol{M}$

Given an input crystal material $\boldsymbol{M}_0 = (\boldsymbol{A}_0, \boldsymbol{X}_0, \boldsymbol{L}_0)$, we define a forward diffusion process through a Markov chain over T steps to defuse $\boldsymbol{A}, \boldsymbol{X}, \boldsymbol{L}$ independently as follows :

$$q(\boldsymbol{A}_t, \boldsymbol{X}_t, \boldsymbol{L}_t | \boldsymbol{A}_{t-1}, \boldsymbol{X}_{t-1}, \boldsymbol{L}_{t-1}) = q(\boldsymbol{A}_t | \boldsymbol{A}_{t-1})q(\boldsymbol{X}_t | \boldsymbol{X}_{t-1})q(\boldsymbol{L}_t | \boldsymbol{L}_{t-1}) \ \ t = 1, 2, ...T \tag{8}$$

## E.1  DIFFUSION ON LATTICE ($\boldsymbol{L}$)

Lattice Matrix $\boldsymbol{L} = [l_1, l_2, l_3]^T \in \mathbb{R}^{3 \times 3}$ is a global feature of the material which determines the shape and symmetry of the unit cell structure. Since $\boldsymbol{L}$ is in continuous space, we leverage the idea of the Denoising Diffusion Probabilistic Model (DDPM) for diffusion on $\boldsymbol{L}$. In specific, given input lattice matrix $\boldsymbol{L}_0 \sim p(\boldsymbol{L})$, the forward diffusion process iteratively diffuses it over T timesteps to a noisy lattice matrix $\boldsymbol{L}_T$ through a transition probability $q(\boldsymbol{L}_t | \boldsymbol{L}_0)$ at each $t^{th}$ step, which can be derived as follows :

$$q(\boldsymbol{L}_t \mid \boldsymbol{L}_0) = \mathcal{N}\left(\boldsymbol{L}_t \mid \sqrt{\bar{\alpha}_t}\boldsymbol{L}_0, \ (1 - \bar{\alpha}_t)\mathbf{I}\right) \tag{9}$$

where, $\bar{\alpha}_t = \prod_{k=1}^{t} \alpha_k$, $\alpha_t = 1 - \beta_t$ and $\{\beta_t \in (0, 1)\}_{t=1}^{T}$ controls the variance of diffusion step following certain variance scheduler. By reparameterization, we can rewrite equation 9 as:

$$\boldsymbol{L}_t = \sqrt{\bar{\alpha}_t}\boldsymbol{L}_0 + \sqrt{1 - \bar{\alpha}_t}\boldsymbol{\epsilon}^{\boldsymbol{L}} \tag{10}$$

where, $\boldsymbol{\epsilon}^l$ is a noise, sampled from $\mathcal{N}(\mathbf{0}, \mathbf{I})$, added with original input sample $\boldsymbol{L}_0$ at $t^{th}$ step to generate $\boldsymbol{L}_t$. After T such diffusion steps, noisy lattice matrix $\boldsymbol{L}_T$ is generated from prior noise distribution $\sim \mathcal{N}(\mathbf{0}, \mathbf{I})$. In the reverse denoising process, given noisy $\boldsymbol{L}_T \sim \mathcal{N}(\mathbf{0}, \mathbf{I})$ we reconstruct true lattice structure $\boldsymbol{L}_0$ thorough iterative denoising step via learning reverse conditional distribution, which we formulate as follows :

$$p(\boldsymbol{L}_{t-1} | \boldsymbol{M}_t, \boldsymbol{C_p}) = \mathcal{N}\left\{\boldsymbol{L}_{t-1} \mid \mu^L(\boldsymbol{M}_t, \boldsymbol{C_p}), \beta_t \frac{(1 - \bar{\alpha}_{t-1})}{(1 - \bar{\alpha}_t)}\mathbf{I}\right\} \tag{11}$$

where $\mu^L(\boldsymbol{M}_t, \boldsymbol{C_p}) = \frac{1}{\sqrt{\alpha_t}}\left(\boldsymbol{L}_t - \frac{1-\alpha_t}{\sqrt{1-\bar{\alpha}_t}} \ \hat{\boldsymbol{\epsilon}}^L(\boldsymbol{M}_t, \boldsymbol{C_p}, t)\right)$. Intuitively, $\hat{\boldsymbol{\epsilon}}^l$ is the denoising term that needs to be subtracted from $\boldsymbol{L}_t$ to generate $\boldsymbol{L}_{t-1}$ and textual representation $\boldsymbol{C_p}$ will steer this reverse diffusion process. We use a text-guided denoising network $\Phi_\theta(\boldsymbol{A}_t, \boldsymbol{X}_t, \boldsymbol{L}_t, t, \boldsymbol{C_p})$ to model the noise

term $\hat{\epsilon}^{\boldsymbol{L}}(\boldsymbol{M}_t, \boldsymbol{C_p}, t)$. Following the simplified training objective proposed by (Ho et al., 2020), we train the aforementioned denoising network using $l_2$ loss between $\hat{\epsilon}^{\boldsymbol{L}}$ and $\epsilon^{\boldsymbol{L}}$

$$\mathcal{L}_{lattice} = \mathbb{E}_{\epsilon^L, t \sim \mathcal{U}(1,T)} \| \boldsymbol{\epsilon^L} - \hat{\boldsymbol{\epsilon}}^{\boldsymbol{L}} \|_2^2 \tag{12}$$

### E.2 DIFFUSION ON ATOM TYPES ($\boldsymbol{A}$)

Prior studies (Jiao et al., 2023; Xie et al., 2021) consider Atom Type Matrix $\boldsymbol{A}$ as the logits/probability distribution for k classes $\in \mathbb{R}^{N \times k}$ (continuous variable in real space) and apply DDPM to learn the distribution. However for discrete data these models are inappropriate and produce suboptimal results (Austin et al., 2021; Campbell et al., 2022; Hoogeboom et al., 2021). Hence we consider $\boldsymbol{A}$ as N discrete variables belonging to k classes and leverage discrete denoising diffusion probabilistic model (D3PM) (Austin et al., 2021) for diffusion on $\boldsymbol{A}$. In specific, denoting row vector $\boldsymbol{a}$ as a one-hot representation of an atom $a$, we can write transition probability for forward process as:

$$q(\boldsymbol{a}_t | \boldsymbol{a}_{t-1}) = Cat(\boldsymbol{a}_t; \boldsymbol{p} = \boldsymbol{a}_{t-1} \boldsymbol{Q}_t) \tag{13}$$

where $Cat(\boldsymbol{a}; \boldsymbol{p})$ is a categorical distribution over the one-hot row vector $\boldsymbol{a}$ with probabilities given by the row vector $\boldsymbol{p}$ and $\boldsymbol{Q}_t$ is the Markov transition matrix at time step t defined as $[\boldsymbol{Q}_t]_{i,j} = q(a_t = i \mid a_{t-1} = j)$. Different choices of $\boldsymbol{Q}_t$ and corresponding stationary distributions are proposed by (Austin et al., 2021) which provides flexibility to control the data corruption and denoising process. We adopted the absorbing state diffusion process, introducing a new absorbing state [MASK] in $\boldsymbol{Q}_t$. At each time step t, we can formally define the transition matrix as:

$$[\boldsymbol{Q}_t]_{i,j} = \begin{cases} 1, & \text{if } i = j = [MASK]. \\ 1 - \beta_t, & \text{if } i = j \neq [MASK] \\ \beta_t, & \text{if } i = j = [MASK]. \end{cases} \tag{14}$$

Intuitively, at each time step t, an atom either stays in its type state with probability $1 - \beta_t$ or moves to [MASK] state with probability $\beta_t$ and once it moves to [MASK] state, it stays in that state. Hence, the stationary distribution of this diffusion process has all the mass on the [MASK] state. During reverse denoising process, given textual representation $\boldsymbol{C_p}$, we first sample noisy $\boldsymbol{a}_T$ and obtain $\boldsymbol{a}_0$ thorough iterative denoising step via learning reverse conditional transition:

$$p_\theta(\boldsymbol{a}_{t-1} | \boldsymbol{a}_t, \boldsymbol{C_p}) \propto \sum_{\boldsymbol{a}_0} q(\boldsymbol{a}_{t-1}, \boldsymbol{a}_t | \boldsymbol{a}_0) p_\theta(\boldsymbol{a}_0 | \boldsymbol{a}_t, \boldsymbol{C_p}) \tag{15}$$

We use the text-guided denoising network $\Phi_\theta(\boldsymbol{A}_t, \boldsymbol{X}_t, \boldsymbol{L}_t, t, \boldsymbol{C_p})$ to model this backward denoising process, which is trained using the following loss function as proposed by (Austin et al., 2021) :

$$\mathcal{L}_{type} = \mathcal{L}_{VB} + \lambda \mathcal{L}_{CE} \tag{16}$$

where $\mathcal{L}_{VB}$ is the variational lower bound loss defined as follows:

$$\mathcal{L}_{VB} = \mathbb{E}_{q(\boldsymbol{a}_0)} \Bigg[ \underbrace{D_{KL}\{q(\boldsymbol{a}_T | \boldsymbol{a}_0) || p(\boldsymbol{a}_T)\}}_{L_T} + \sum_{t=2}^{T} \mathbb{E}_{q(\boldsymbol{a}_t | \boldsymbol{a}_0)} \underbrace{[D_{KL}\{q(\boldsymbol{a}_{t-1} | \boldsymbol{a}_t, \boldsymbol{a}_0) || p_\theta(\boldsymbol{a}_{t-1} | \boldsymbol{a}_t)\}]}_{L_{t-1}}$$

$$- \underbrace{\mathbb{E}_{q(\boldsymbol{a}_1 | \boldsymbol{a}_0)}[\log p_\theta(\boldsymbol{a}_0 | \boldsymbol{a}_1)]}_{L_0} \Bigg]$$

$$\tag{17}$$

and $\mathcal{L}_{CE}$ is the cross-entropy loss defined as follows:

$$\mathcal{L}_{CE} = \mathbb{E}_{q(\boldsymbol{a}_0)} \Bigg[ \sum_{t=2}^{T} \mathbb{E}_{q(\boldsymbol{a}_t | \boldsymbol{a}_0)}[\log p_\theta(\boldsymbol{a}_0 | \boldsymbol{a}_t)\}] \Bigg] \tag{18}$$

and $\lambda$ is a hyperparameter.

### E.3 DIFFUSION ON ATOM COORDINATES ($\boldsymbol{X}$)

Coordinate Matrix $\boldsymbol{X} = [\boldsymbol{x}_1, \boldsymbol{x}_2, ..., \boldsymbol{x}_N]^T \in \mathbb{R}^{N \times 3}$ denotes atomic coordinate positions, where $x_i \in \mathbb{R}^3$ corresponds to coordinates of $i^{th}$ atom in the unit cell. We can diffuse the atom coordinates in two ways: either by diffusing cartesian coordinates or fractional coordinates. Prior works like CDVAE (Xie et al., 2021) and SyMat (Luo et al., 2023b) diffuse cartesian coordinates whereas DiffCSP (Jiao et al., 2023) diffuse fractional coordinates. In our setup, as we are jointly learning atom coordinates and lattice matrix simultaneously, we follow the line of work by DiffCSP and diffuse fractional coordinates. Atomic fractional coordinates in crystal material lives in quotient space $\mathbb{R}^{N \times 3} / \mathbb{Z}^{N \times 3}$ induced by the crystal periodicity. Since the Gaussian distribution used in DDPM is unable to model the cyclical and bounded domain of $\boldsymbol{X}$, it is not suitable to apply DDPM to model $\boldsymbol{X}$. Hence at each step of forward diffusion, we add noise sample from Wrapped Normal

(WN) distribution (De Bortoli et al., 2022) to $\boldsymbol{X}$ and during backward diffusion leverage Score Matching Diffusion Networks (Song & Ermon, 2019; 2020) to model underlying transition probability $q(\boldsymbol{X}_t \mid \boldsymbol{X}_0) = \mathcal{N}_W(\boldsymbol{X}_t \mid \boldsymbol{X}_0, \sigma_t^2 \mathbf{I})$. In specific, at each $t^{th}$ step of diffusion, we derive $\boldsymbol{X}_t$ as : $\boldsymbol{X}_t = f_w(\boldsymbol{X}_0 + \boldsymbol{\sigma_t}\boldsymbol{\epsilon}^X)$ where, $\boldsymbol{\epsilon}^X$ is a noise, sampled from $\mathcal{N}(\boldsymbol{0}, \mathbf{I})$, $\boldsymbol{\sigma_t}$ is the noise scale following exponential scheduler and $f_w(.)$ is a truncation function. Given a fractional coordinate matrix X, truncation function $f_w(\boldsymbol{X}) = (\boldsymbol{X} - \lfloor \boldsymbol{X} \rfloor)$ returns the fractional part of each element of $\boldsymbol{X}$.

As argued in (Jiao et al., 2023), $q(X_t|X_0)$ is periodic translation equivariant, and approaches uniform distribution $\mathcal{U}(0, 1)$ for sufficiently large values of $\sigma_T$. Hence during the backward denoising process, we first sample $X_T \sim \mathcal{U}(0, 1)$ and iteratively denoise via score network for T steps to recover back the true fractional coordinates $X_0$. We use the text-guided denoising network $\Phi_\theta(\boldsymbol{A}_t, \boldsymbol{X}_t, \boldsymbol{L}_t, t, \boldsymbol{C_p})$ to model the backward diffusion process, which is trained using the following score-matching objective function :

$$\mathcal{L}_{coord} = \mathbb{E}_{\substack{\boldsymbol{X}_t \sim q(\boldsymbol{X}_t|\boldsymbol{X}_0) \\ t \sim \mathcal{U}(1,T)}} \|\nabla_{\boldsymbol{X}_t} \log q(\boldsymbol{X}_t|\boldsymbol{X}_0) - \hat{\boldsymbol{\epsilon}}^X(\boldsymbol{M}_t, \boldsymbol{C_p}, t)\|_2^2 \tag{19}$$

where $\nabla_{\boldsymbol{X}_t} \log q(\boldsymbol{X}_t|\boldsymbol{X}_0) \propto \sum_{\boldsymbol{K} \in \mathbb{Z}^{N \times 3}} \exp(-\frac{\|\boldsymbol{X}_t - \boldsymbol{X}_0 + \boldsymbol{K}\|_F^2}{2\sigma_t^2})$ is the score function of transitional distribution and $\hat{\boldsymbol{\epsilon}}^X(\boldsymbol{M}_t, \boldsymbol{C_p}, t)$ denoising term.

### E.4 TEXT GUIDED DENOISING NETWORK

In this subsection, we will illustrate the detailed architecture of our proposed Text Guided Denoising Network $\Phi_\theta(\boldsymbol{A}_t, \boldsymbol{X}_t, \boldsymbol{L}_t, t, \boldsymbol{C_p})$, which we used to denoise $\boldsymbol{A}$, $\boldsymbol{X}$ and $\boldsymbol{L}$. As mentioned in 2, the learned distribution of material structure $p(\boldsymbol{M})$ must satisfy periodic E(3) invariance. Hence we leverage an periodic-E(3)-equivariant Graph Neural Network (GNN) integrated with a pre-trained textual encoder to model the denoising process. In particular, as a text encoder, we adopt a pre-trained MatSciBERT (Gupta et al., 2022) model, which is a domain-specific language model for materials science, followed by a projection layer. MatSciBERT is effectively a pre-trained SciBERT model on a scientific text corpus of 3.17B words, which is further trained on a huge text corpus of materials science containing around 285 M words. We feed textual description of material $\mathcal{T}$ and extract embedding of [CLS] token $\boldsymbol{h}_{CLS}$ as a representation of the whole text. Further. we pass $\boldsymbol{h}_{CLS}$ through a projection layer to generate the contextual textual embedding for the material $\boldsymbol{C_p} \in \mathbb{R}^d$, which we pass to the equivariant GNN model to guide the denoising process. Practically, as the backbone network for the backward diffusion process, we extend CSPNet architecture (Jiao et al., 2023), originally developed for crystal structure prediction (CSP) task. CSPNet is built upon EGNN (Satorras et al., 2021), satisfying periodic E(3) invariance condition on periodic crystal structure. At the $k^{th}$ layer message passing, the Equivariant Graph Convolutional Layer (EGCL) takes as input the set of atom embeddings $\boldsymbol{h}^k = [\boldsymbol{h}_1^k, \boldsymbol{h}_2^k, ..., \boldsymbol{h}_N^k]$, atom coordinates $\boldsymbol{x}^k = [\boldsymbol{x}_1^k, \boldsymbol{x}_2^k, ..., \boldsymbol{x}_N^k]$ and Lattice Matrix $\boldsymbol{L}$ and outputs a transformation on $\boldsymbol{h}^{k+1}$. Formally, we can define the $k^{th}$ layer message passing operation as follows :

$$\boldsymbol{m}_{i,j} = \rho_m\{\boldsymbol{h}_i^k, \ \boldsymbol{h}_j^k, \ \boldsymbol{L}^T\boldsymbol{L}, \ \psi_{FT}(\boldsymbol{x}_i^k - \boldsymbol{x}_j^k)\}; \tag{20}$$

$$\boldsymbol{h}_i^{k+1} = \boldsymbol{h}_i^k + \rho_h\{\boldsymbol{h}_i^k, \boldsymbol{m}_i\} \tag{21}$$

where $\boldsymbol{m}_i = \sum_{j=1}^N \boldsymbol{m}_{i,j}$, $\rho_m, \rho_h$ are multi-layer perceptrons and $\psi_{FT}$ is a Fourier Transformation function applied on relative difference between fractional coordinates $\boldsymbol{x}_i^k, \boldsymbol{x}_j^k$. Fourier Transformation is used since it is invariant to periodic translation and extracts various frequencies of all relative fractional distances that are helpful for crystal structure modeling.

We fuse textual representation $\boldsymbol{C_p}$ into input atom feature $\boldsymbol{h}_i^0$ as

$$\boldsymbol{h}_i^0 = \rho \ \{ \ f_{atom}(\boldsymbol{a}_i) \ || \ f_{pos}(t) \ || \ \boldsymbol{C_p} \tag{22}$$

where t is the timestamp of the diffusion model, $f_{pos}(.)$ is sinusoidal positional encoding (Ho et al., 2020; Vaswani et al., 2017), $f_{atom}(.)$ learned atomic embedding function and $||$ is concatenation operation. Input atom features $\boldsymbol{h}^0$ and coordinates $\boldsymbol{x}^0$ are fed through $\mathcal{K}$ layers of EGCL to produce $\hat{\boldsymbol{\epsilon}}^L$, $p(\boldsymbol{A}_{t-1} \mid \boldsymbol{M}_t)$ and $\hat{\boldsymbol{\epsilon}}^X$ as follows :

$$\hat{\boldsymbol{\epsilon}}^L = \boldsymbol{L}\rho_L(\frac{1}{N}\sum_N^{i=1} \boldsymbol{h}^{\mathcal{K}});$$

$$p(\boldsymbol{A}_{t-1} \mid \boldsymbol{M}_t) = \rho_A(\boldsymbol{h}^{\mathcal{K}}); \tag{23}$$

$$\hat{\boldsymbol{\epsilon}}^X = \rho_X(\boldsymbol{h}^{\mathcal{K}})$$

where $\rho_L, \rho_A, \rho_X$ are multi-layer perceptrons on the final layer embeddings. Intuitively, we feed global structural knowledge about the crystal structure into the network by injecting contextual representation $C_p$ into input atom features. This added signal will participate through message-passing operations in Eq. 20 and guides in denoising atom types, coordinates, and lattice parameters such that it can capture the global crystal geometry and aligned with the input stable structure specified by textual description.

---

**Algorithm 1** Training Algorithm

---

1: **Input:** Atom type Matrix $A_0$ (One hot Vector Representation), Coordinate Matrix $X_0$, Lattice matrix $L_0$, Markov Transition Matrix $[Q_t]_{t=1}^T$, Textual Representation $C_p$, Number of diffusion step T and hyperparameters $\lambda_A, \lambda_X, \lambda_L$.
2: **repeat**
3:     Sample $t \sim \mathcal{U}(\mathbf{0}, \mathbf{T})$
4:     Sample Noise $\epsilon^{\mathbf{X}}, \epsilon^{\mathbf{L}} \sim N(0, I)$
5:     $L_t = \sqrt{\bar{\alpha}_t} L_0 + \sqrt{1 - \bar{\alpha}_t} \epsilon^L$
6:     $X_t = f_w(X_0 + \sigma_t \epsilon^x)$
7:     $A_t = Cat(A_t; p = A_{t-1} Q_t)$
8:     $\hat{\epsilon}^{\mathbf{L}}, \hat{\epsilon}^{\mathbf{X}}, A'_t \leftarrow \Phi_\theta(\mathbf{A}_t, \mathbf{X}_t, \mathbf{L}_t, t, \mathbf{C_p})$
9:     $\mathcal{L}_{lattice} = \| \epsilon^L - \hat{\epsilon}^L \|_2^2$
10:    $\mathcal{L}_{coord} = \| \nabla_{X_t} \log q(X_t | X_0) - \hat{\epsilon}^X \|_2^2$
11:    $\mathcal{L}_{type} = \mathcal{L}_{VB} + \lambda \mathcal{L}_{CE}$
12:    Minimize $\mathcal{L} = \lambda_L \mathcal{L}_{lattice} + \lambda_A \mathcal{L}_{type} + \lambda_X \mathcal{L}_{coord}$ and update parameters of $\Phi_\theta$
13: **until** converged

---

---

**Algorithm 2** Sampling Algorithm

---

1: Sample $L_T \sim \mathcal{N}(\mathbf{0}, \mathbf{I}), X_T \sim \mathcal{U}(0, 1)$
2: Randomly sample each atom type between 0 to 99 (Max possible atom type) and form $A_T$
3: $C_p \leftarrow$ Textual Representation
4: **for** $t \leftarrow T$ to 1 **do**
5:     $\epsilon^{\mathbf{A}}, \epsilon^{\mathbf{X}}, \epsilon^{\mathbf{L}} \sim N(0, I) /* Sample */$
6:     $\hat{\mathbf{A}}, \hat{\epsilon}^{\mathbf{X}}, \hat{\epsilon}^{\mathbf{L}} \leftarrow \Phi_\theta(\mathbf{A}_t, \mathbf{X}_t, \mathbf{L}_t, t, \mathbf{C_p})$
7:     $\mathbf{L}_{t-1} \leftarrow \frac{1}{\sqrt{\alpha_t}} (\mathbf{L}_t - \frac{\beta_t}{\sqrt{1-\bar{\alpha}_t}} \hat{\epsilon}^{\mathbf{L}}) + \sqrt{\beta_t \frac{1-\bar{\alpha}_{t-1}}{1-\bar{\alpha}_t}} \epsilon^{\mathbf{L}}$
8:     $\mathbf{A}_{t-1} \leftarrow \text{Softmax}(\hat{\mathbf{A}} + \sigma_t \epsilon^{\mathbf{A}})$
9:     $\mathbf{X}_{t-\frac{1}{2}} \leftarrow w(\mathbf{X}_t + (\sigma_t^2 - \sigma_{t-1}^2)\hat{\epsilon}^{\mathbf{X}} + \frac{\sigma_{t-1}\sqrt{\sigma_t^2 - \sigma_{t-1}^2}}{\sigma_t}\epsilon^{\mathbf{X}})$
10:    $, \hat{\epsilon}^{\mathbf{X}} \leftarrow \Phi_\theta(\mathbf{A}_t, \mathbf{X}_{t-\frac{1}{2}}, \mathbf{L}_{t-1}, t, \mathbf{C_p})$
11:    $\eta_t \leftarrow step\_size * \frac{\sigma_{t-1}}{\sigma_t}$
12:    $\mathbf{X}_{t-1} \leftarrow w(\mathbf{X}_{t-\frac{1}{2}} + \eta_t \hat{\epsilon}^{\mathbf{X}} + \sqrt{2\eta_t}\epsilon^{\mathbf{X}})$
13: **end for**

---

### E.5 TRAINING AND SAMPLING

TGDMat is trained using the following combined loss:

$$\mathcal{L} = \lambda_L \mathcal{L}_{lattice} + \lambda_A \mathcal{L}_{type} + \lambda_X \mathcal{L}_{coord} \tag{24}$$

where $\mathcal{L}_{lattice}, \mathcal{L}_{type}$ and $\mathcal{L}_{coord}$ are lattice $l_2$ loss (Eq. 12), type cross-entropy loss (Eq. 16) and coordinate score matching loss (Eq. 19) respectively and $\lambda_L, \lambda_A, \lambda_X$ are hyperparameters control the relative weightage between these different loss components. During training, we freeze the MatSciBERT parameters and do not tune it further. During sampling, we use the Predictor-Corrector sampling mechanism to sample $A_0, X_0$ and $L_0$. Next we explain algorithms for training and sampling.

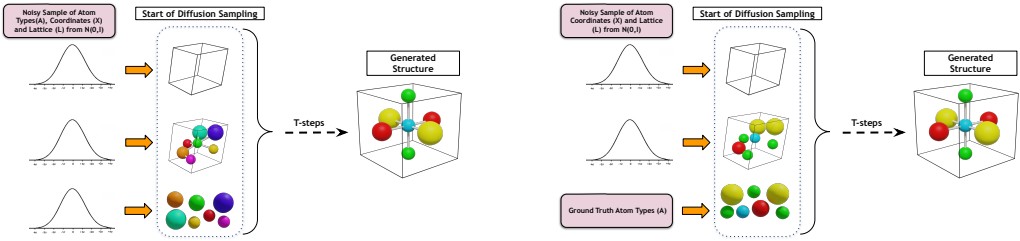

(a) Random Material Generation (Gen) Task      (b) Crystal Structure Prediction (CSP) Task

Figure 5

# F   EXPERIMENTS

## F.1   EXPERIMENTAL SETUP

**Benchmark Tasks.** We evaluate our proposed model TGDMat on two different categories of tasks for material generation, *Random Material Generation (Gen)* and *Crystal Structure Prediction (CSP)*. In *Gen* task, the goal of the generative model is to generate novel stable materials (atom types, fractional coordinates, and lattice structure). In *CSP* task, atom types of the materials are given and the goal is to predict/match the crystal structure (atom coordinates and lattice). In TGDMat model, by design choice, we use the textual description of crystal materials during each step of the reverse diffusion process to enhance the generation capability in both tasks. A pictorial illustration of both tasks is provided at 5

**Dataset.** Following Xie et al  (Xie et al., 2021) we evaluate our model on three baseline datasets: **Perov-5**, **Carbon-24** and **MP-20**. **Perov-5**  (Castelli et al., 2012a;b) dataset consists of 18,928 perovskite materials, each with 5 atoms in a cell. They generally can be denoted by **ABX₃** indicating the three different types of atoms usually observed in such materials. **Carbon-24** (Pickard., 2020) dataset has 10,153 materials with 6 to 24 atoms of carbon in the crystal lattice. Finally, **MP-20** (Jain et al., 2013b) dataset has 45,231 materials curated from the Materials Project library  (Jain et al., 2013a), where each material has at most 20 atoms in the lattice. Crystals from **Perov-5** dataset share the same structure but differ in composition, whereas Crystals from **Carbon-24** share the same composition but differ in structure. Crystals from **MP-20** differs in both structure and composition. We curated textual data for these datasets with a textual description of each material. Specifically, we generate both long detailed textual descriptions and shorter prompts using approaches mentioned in Appendix D.
The structures in all three datasets are derived from quantum mechanical simulations and are all at local energy minima. Most materials in **Perov-5** and **Carbon-24** are hypothetical, whereas **MP-20** represents a realistic dataset that includes many experimentally known inorganic materials, each with a maximum of 20 atoms in the unit cell, most of which are globally stable. A model that performs well on MP-20 could potentially generate novel materials that can be synthesized experimentally. While training TGDMat, we split the datasets into the train, test, and validation sets following the convention of 60:20:20 as done by Xie et al  (Xie et al., 2021).

**Hyper-Parameters Details.** In our TGDMat model, we adopted 4 layers CSPNet as message passing layer with hidden dimension set as 512. Further, we use pre-trained MatSciBERT (Gupta et al., 2022) followed by a two-layer projection layer (projection dimension 64) as the text encoder module. We keep the dimension of time embedding at each diffusion timestep as 64. We train it for 500 epochs using the same optimizer, and learning rate scheduler as DiffCSP and keep the batch size as 512. We perform all the experiments in the Tesla P100-PCIE-16GB GPU server.

## F.2 Evaluation Metrics

**Random Material Generation (Gen) Task.** Following CDVAE (Xie et al., 2021), we evaluate the performance of TGDMat and baseline models on generating novel material structure using seven metrics under three broad categories: **Validity**, **Coverage**, and **Property Statistics**. Under **Validity**, following the prior line of work (Court et al., 2020; Xie et al., 2021), we measure structural and compositional validity, representing the percentages of generated crystals with valid periodic structures and atom types, respectively. A structure is valid as long as the shortest distance between any pair of atoms is larger than 0.5 Å whereas the composition is valid if the overall charge is neutral as computed by SMACT (Davies et al., 2019). In **Coverage**, we consider two coverage metrics, COV-R (Recall) and COV-P (Precision). COV-R measures the percentage of the test set materials being correctly predicted, whereas COV-P measures the percentage of generated materials that cover at least one of the test set materials. (More detailed discussions can be found in (Xie et al., 2021) and (Ganea et al., 2021)). Finally, we evaluate the similarity between the generated materials and those in the test set using various **Property Statistics**, where we compute the earth mover's distance (EMD) between the distributions in element number (# Elem), density ($\rho$, unit g/cm3), and formation energy ($\mathcal{E}$, unit eV/atom) predicted by a GNN model.

**Crystal Structure Prediction (CSP) Task.** We evaluate the performance of TGDMat and baseline models on stable structure prediction using standard metrics proposed by the prior works (Jiao et al., 2023; Xie et al., 2021), by matching the generated structure and the input ground truth structure in the test set. In Specific, for each material structure in the test set, we generate k samples given the textual description and then identify the matching if at least one of the samples matches the ground truth structure. We calculate the **Match Rate** and **RMSE** metrics using the StructureMatcher class in Pymatgen, which identifies the best match between two structures while accounting for all material invariances. Match rate indicates the percentage of the matched structures over the test set satisfying thresholds stol=0.5, angle_tol=10, ltol=0.3. RMSE is computed between the ground truth and the best-matching candidate, normalized by $\sqrt[3]{V/N}$ where V is the volume of the lattice, and averaged over the matched structures. For baselines and TGDMat, we evaluate using $k = 1$ and $k = 20$.

## F.3 Complete and Detailed Results

In this subsection, we provide full comprehensive results on both Gen and CSP tasks across three benchmark datasets and evaluate the performance of all the baseline models, their text-guided variants (both short and long), and our proposed TGDMat(Long) & TGDMat(Short). We report the CSP and Gen task results in Table 6 and 7 respectively.

Following are the Insights or Observations:

- For both tasks, across all the datasets, text guidance outperforms the vanilla diffusion models in almost all metrics.

- Our experiments suggest that using shorter prompts text-guided models outperforms the vanilla baseline models. However, performance is even superior when using text-guided diffusion using longer prompts.

- For the CSP task, using text guidance during the reverse denoising process, with just one generated sample per test material, text-guided variants outperform respective vanilla models, thereby reducing computational overhead.

- Our proposed TGDMat (Long) stands out as the leading model when compared to all baseline models and their text-guided variants across three benchmark datasets. In specific, for Gen Task, TGDMat (Long) outperforms the closest baseline DiffCSP+ (Long) because we leveraged discrete diffusion on atom types, which is more powerful in learning discrete variables like atom types.

- Finally, results indicate that utilizing shorter prompts TGDMat (Short) results in a slight decrease in overall performance compared to the longer variant TGDMat (Long). Nonetheless, the performance remains superior or comparable to baseline models (vannila and text-guided variants).

| Method | # samples | Perov-5 | | Carbon-24 | | MP-20 | |
|---|---|---|---|---|---|---|---|
| | | Match | RMSE | Match | RMSE | Match | RMSE |
| CDVAE | 1 | 45.31 | 0.1138 | 17.09 | 0.2969 | 33.9 | 0.1045 |
| | 20 | 88.51 | 0.0464 | 88.37 | 0.2286 | 66.95 | 0.1026 |
| CDVAE+(short) | 1 | 48.97 | 0.1063 | 22.65 | 0.264 | 40.33 | 0.1037 |
| | 20 | 89.54 | 0.0423 | 89.61 | 0.2188 | 70.22 | 0.0876 |
| CDVAE+(long) | 1 | 49.25 | 0.1055 | 23.73 | 0.259 | 41.8 | 0.1021 |
| | 20 | 89.73 | 0.0417 | 89.77 | 0.2053 | 72.56 | 0.084 |
| SyMat | 1 | 47.32 | 0.1074 | 20.81 | 0.2655 | 33.92 | 0.1039 |
| | 20 | 90.25 | 0.0316 | 89.29 | 0.2184 | 71.03 | 0.0945 |
| SyMat+(short) | 1 | 49.39 | 0.0985 | 23.71 | 0.2567 | 40.84 | 0.1027 |
| | 20 | 92.1 | 0.0255 | 90.86 | 0.2069 | 71.31 | 0.0875 |
| SyMat+(long) | 1 | 50.88 | 0.0963 | 28.18 | 0.251 | 43.17 | 0.1016 |
| | 20 | 92.3 | 0.0201 | 91.65 | 0.187 | 72.96 | 0.082 |
| DiffCSP | 1 | 52.02 | 0.076 | 17.54 | 0.2759 | 51.49 | 0.0631 |
| | 20 | 98.6 | 0.0128 | 88.47 | 0.2192 | 77.93 | 0.0492 |
| DiffCSP+(short) | 1 | 56.54 | 0.0583 | 24.13 | 0.2424 | 52.22 | 0.0597 |
| | 20 | 98.25 | 0.0137 | 88.28 | 0.2252 | 80.97 | 0.0443 |
| DiffCSP+(long) | 1 | 90.46 | 0.0203 | 44.63 | 0.2266 | 55.15 | 0.0572 |
| | 20 | 98.59 | 0.0072 | 95.27 | 0.1534 | 82.02 | 0.0391 |
| TGDMat (short | 1 | 56.54 | 0.0583 | 24.13 | 0.2424 | 52.22 | 0.0597 |
| | 20 | 98.25 | 0.0137 | 88.28 | 0.2252 | 80.97 | 0.0443 |
| TGDMat (long) | 1 | 90.46 | 0.0203 | 44.63 | 0.2266 | 55.15 | 0.0572 |
| | 20 | 98.59 | 0.0072 | 95.27 | 0.1534 | 82.02 | 0.0391 |

Table 6: Summary of the Complete and Detailed Results on the CSP Task.

| Dataset | Method | Validity | | Coverage | | Property | | |
|---------|--------|----------|--------|-------|-------|-----------|---------|-------------|
| | | Comp | Struct | Cov-R | Cov-P | # element | Density | form_energy |
| Perov 5 | CDVAE | 98.29 | 100 | 99.25 | 98.39 | 0.0731 | 0.1462 | 0.0291 |
| | CDVAE+ (Short) | 98.17 | 100 | 99.4 | 99.01 | 0.0706 | 0.1395 | 0.0246 |
| | CDVAE+ (Long) | 98.45 | 100 | 99.53 | 99.09 | 0.0609 | 0.1276 | 0.0223 |
| | SyMat | 96.83 | 100 | 99.16 | 98.29 | 0.0193 | 0.1991 | 0.2827 |
| | SyMat+ (Short) | 96.94 | 100 | 99.22 | 98.4 | 0.0192 | 0.1827 | 0.2633 |
| | SyMat+ (Long) | 97.88 | 100 | 99.71 | 98.79 | 0.0172 | 0.1755 | 0.2566 |
| | DiffCSP | 98.15 | 100 | 99.28 | 98.08 | 0.0132 | 0.1281 | 0.0267 |
| | DiffCSP+ (Short) | 98.21 | 100 | 99.61 | 98.39 | 0.0123 | 0.1193 | 0.0266 |
| | DiffCSP+ (Long) | 98.44 | 100 | 99.85 | 98.53 | 0.0119 | 0.1071 | 0.0241 |
| | TGDMat(Short) | 98.28 | 100 | 99.71 | 99.24 | 0.0108 | 0.0947 | 0.0237 |
| | TGDMat(Long) | 98.63 | 100 | 99.87 | 99.52 | 0.009 | 0.0497 | 0.0187 |
| Carbon 24 | CDVAE | - | 100 | 99.35 | 82.66 | - | 0.1539 | 0.2889 |
| | CDVAE+ (Short) | - | 100 | 99.34 | 82.96 | - | 0.1398 | 0.2804 |
| | CDVAE+ (Long) | - | 100 | 99.82 | 84.76 | - | 0.1377 | 0.266 |
| | SyMat | - | 100 | 99.42 | 97.17 | - | 0.1234 | 3.9628 |
| | SyMat+ (Short) | - | 100 | 99.52 | 97.2 | - | 0.1206 | 3.7422 |
| | SyMat+ (Long) | - | 100 | 99.9 | 97.63 | - | 0.1171 | 3.862 |
| | DiffCSP | - | 99.9 | 99.49 | 97.27 | - | 0.0861 | 0.0876 |
| | DiffCSP+ (Short) | - | 100 | 99.61 | 97.29 | - | 0.0811 | 0.087 |
| | DiffCSP+ (Long) | - | 100 | 99.93 | 97.33 | - | 0.0763 | 0.0853 |
| | TGDMat(Short) | - | 100 | 99.81 | 91.77 | - | 0.0681 | 0.0865 |
| | TGDMat(Long) | - | 100 | 99.91 | 92.43 | - | 0.0436 | 0.0632 |
| MP 20 | CDVAE | 86.3 | 100 | 99.15 | 99.49 | 1.4921 | 0.7085 | 0.3039 |
| | CDVAE+ (Short) | 87.05 | 100 | 99.36 | 99.6 | 0.993 | 0.642 | 0.297 |
| | CDVAE+ (Long) | 87.42 | 100 | 99.57 | 99.81 | 0.972 | 0.6388 | 0.2977 |
| | SyMat | 87.96 | 99.9 | 98.3 | 99.37 | 0.5236 | 0.4012 | 0.3877 |
| | SyMat+ (Short) | 88.08 | 99.9 | 98.59 | 99.47 | 0.5031 | 0.3917 | 0.3622 |
| | SyMat+ (Long) | 88.47 | 99.9 | 99.01 | 99.95 | 0.4865 | 0.3879 | 0.3489 |
| | DiffCSP | 83.25 | 100 | 99.41 | 99.76 | 0.3411 | 0.3802 | 0.1497 |
| | DiffCSP+ (Short) | 84.57 | 100 | 99.52 | 99.85 | 0.331 | 0.38 | 0.1379 |
| | DiffCSP+ (Long) | 85.07 | 100 | 99.81 | 99.89 | 0.3122 | 0.3799 | 0.1355 |
| | TGDMat(Short) | 86.6 | 100 | 99.79 | 99.88 | 0.3337 | 0.3296 | 0.1189 |
| | TGDMat(Long) | 92.97 | 100 | 99.89 | 99.95 | 0.289 | 0.3082 | 0.1154 |

Table 7: Summary of the Complete and Detailed Results on the Gen Task.

## F.4   Correctness of Generated Materials

**Setup.** In this section, we investigate whether the generated material matches different features specified by the textual prompts. TGDMat has the capability to process textual prompts given by the user, enabling it to manage global attributes about crystal materials such as Formula, Space group, Crystal System, and different property values like formation energy, band-gap, etc. To ensure the fidelity of our model's outputs concerning these specified global attributes from the text prompt, We randomly generated 1000 materials (sampled from all three Datasets) based on their respective textual descriptions(both Long and Short) and assessed the percentage of generated materials that matched the global features outlined in the text prompt. In specific, we matched the Formula, Space group, and Crystal System, and Dimensions of generated materials with the textual descriptions. Moreover, we examined whether properties such as formation energy and bandgap matched the specified criteria as per the text prompt (positive/negative, zero/nonzero).

**Results and Discussions.** We report the results in Table 8. In general, using longer text, considering Perov-5 and Carbon-24 datasets, the generated material meets the specified criteria effectively. However, when dealing with the MP-20 dataset, which is more intricate due to its complex structure and composition, performance tends to decline. Additionally, when using shorter prompts, overall performance suffers across all datasets compared to longer text inputs. This is because the longer text, provided by the robocrystallographer, offers a comprehensive range of information, both global and local, thereby enhancing the generation capabilities of TGDMat.

| Method | Global Features in Text Prompt | % of Matched Materials | | |
|---|---|---|---|---|
| | | Perov-5 | Carbon-24 | MP-20 |
| TGDMat(Long) | Formula | 97.50 | 98.20 | 70.54 |
| | Space Group | 87.00 | 80.79 | 67.88 |
| | Crystal System | 92.60 | 91.55 | 73.54 |
| | Formation Energy | 95.49 | - | 92.88 |
| | Band Gap | - | 98.61 | 96.73 |
| TGDMat(Long) | Formula | 90.70 | 92.56 | 65.22 |
| | Space Group | 86.51 | 80.50 | 58.77 |
| | Crystal System | 83.19 | 81.64 | 72.77 |
| | Formation Energy | 90.33 | - | 91.00 |
| | Band Gap | - | 95.90 | 93.33 |

Table 8: Summary of results on % of generated materials matching different global features specified by the textual prompts.

## F.5   Performance on More Shorter Prompts

In this section, we explore the generalizability and robustness of our model by examining potential variability in text description lengths. The goal of this paper is, given the text prompt, to generate specific material, not any generic or class of materials. Hence some minimum essential information about the crystal, like formula, space group, crystal system, property value, etc must be given as input to the pre-trained model. However, to investigate the robustness of our proposed TGDMat model with more custom and shorter prompts, we did an experiment where we evaluated TGDMat (trained with full text) with even shorter custom prompts with very little information as follows:

- **Specifying only Formula:** *"The chemical formula is GaSiSO2. The elements are Ga, Si, S, O. Generate the material."*

- **Specifying only Space Group Info:** *"The spacegroup number is 1. Generate the material."*

- **Specifying only Property Info:** *"The formation energy per atom is positive. Generate the material."*

We report the results in table 9. We observe that though TGDMat can handle more custom prompts, but it affects the quality of generated materials. Hence we conclude some minimum essential information about the crystal must be given as input to TGDMat to generate high quality crystal materials.

| Text Encoder | Perov-5 | | Carbon-24 | | MP-20 | |
|---|---|---|---|---|---|---|
| | Comp(%) ↑ | Struct (%) ↑ | Comp(%) ↑ | Struct (%) ↑ | Comp(%) ↑ | Struct (%) ↑ |
| Only Formula | 97.06 | 99.19 | - | 98.76 | 86.16 | 96.01 |
| Only Space Group | 85.91 | 98.97 | - | 95.39 | 84.22 | 96.88 |
| Only Property | 96.62 | 98.53 | - | 94.21 | 86.53 | 91.73 |
| Full Text | **98.28** | **100** | - | **100** | **86.60** | **100** |

Table 9: Summary of results on generated materials using more custom/shorter Prompt.

Table 10: Ablation study results on different choices of Text Encoders.

| Text Encoder | Perov-5 | | Carbon-24 | | MP-20 | |
|---|---|---|---|---|---|---|
| | MR ↑ | RMSE ↓ | MR ↑ | RMSE ↓ | MR ↑ | RMSE ↓ |
| BERT | 96.64 | 0.0109 | 72.21 | 0.2679 | 79.53 | 0.057 |
| MatSciBERT | **98.63** | **0.0072** | **95.27** | **0.1534** | **82.02** | **0.039** |
| | Comp ↑ | Struct ↑ | Comp ↑ | Struct ↑ | Comp ↑ | Struct ↑ |
| BERT | 97.44 | 99.97 | - | **100** | 84.73 | 98.37 |
| MatSciBERT | **98.63** | **100** | - | **100** | **92.97** | **100** |

## F.6 ABLATION STUDY : CHOICE OF TEXT ENCODER

Further, we investigate the expressiveness of textual representation during the reverse diffusion process. In particular, we are interested in understanding whether there are any benefits we are gaining from using a domain-specific pre-trained text encoder MatSciBERT. We conduct an ablation study where we substitute MatSciBERT with pre-trained BERT (Devlin et al., 2018) model (which is domain agnostic) as text encoder in TGDMat and evaluate the performance on both tasks. The results presented in Table 10 demonstrate that MatSciBERT surpasses BERT (Devlin et al., 2018) in performance for both tasks. This highlights the richer expressiveness of contextual representation achieved through the use of a domain-specific pre-trained language model.

## F.7 MORE VISUALIZATION ON **PEROV-5**, **CARBON-24** AND **MP-20**

| Detailed Description | Short Prompt | Ground truth | Generated Samples |
|---|---|---|---|
| YCoSO2 crystallizes in the orthorhombic Pmm2 space group. Y is bonded in a distorted square co-planar geometry to two equivalent S, two equivalent O, and two equivalent O atoms. Both Y-S bond lengths are 2.74 Å. Both Y-O bond lengths are 2.24 Å. There is one shorter (2.09 Å) and one longer (2.39 Å) Y-O bond length. Co is bonded in a distorted see-saw-like geometry to two equivalent S and two equivalent O atoms. Both Co-S bond lengths are 2.31 Å. Both Co-O bond lengths are 2.40 Å. S is bonded in a 6-coordinate geometry to two equivalent Y, two equivalent Co, and two equivalent O atoms. Both S-O bond lengths are 2.30 Å. There are two inequivalent O sites. In the first O site, O is bonded in a rectangular see-saw-like geometry to two equivalent Y and two equivalent Co atoms. In the second O site, O is bonded in a distorted square co-planar geometry to two equivalent Y and two equivalent S atoms. | Below is a description of a bulk material. The chemical formula is YCoSO2. The elements are Y, Co, S, O. The formation energy per atom is positive. The space-group number is 24. The crystal system is orthorhombic. Generate the material: |  |  |
| ScMoN2O is (Cubic) Perovskite-derived structured and crystallizes in the tetragonal P4mm space group. Sc is bonded to four equivalent N and two equivalent O atoms to form ScN4O2 octahedra that share corners with six equivalent ScN4O2 octahedra and faces with eight equivalent MoN8O4 cuboctahedra. The corner-sharing octahedral tilt angles range from 0-1°. All Sc-N bond lengths are 2.00 Å. There is one shorter (2.00 Å) and one longer (2.01 Å) Sc-O bond length. Mo is bonded to eight equivalent N and four equivalent O atoms to form MoN8O4 cuboctahedra that share corners with twelve equivalent MoN8O4 cuboctahedra, faces with six equivalent MoN8O4 cuboctahedra, and faces with eight equivalent ScN4O2 octahedra. There are four shorter (2.83 Å) and four longer (2.84 Å) Mo-N bond lengths. All Mo-O bond lengths are 2.83 Å. N is bonded in a linear geometry to two equivalent Sc and four equivalent Mo atoms. O is bonded in a linear geometry to two equivalent Sc and four equivalent Mo atoms. The formation energy per atom is 1.8931. | Below is a description of a bulk material. The chemical formula is ScMoN2O. The elements are Sc, Mo, N, O. The formation energy per atom is positive. The spacegroup number is 98. The crystal system is tetragonal. Generate the material: |  |  |
| ScNO2Ga is alpha Rhenium trioxide-derived structured and crystallizes in the orthorhombic Pmm2 space group. The structure consists of one Ga cluster inside a ScNO2 framework. In the Ga cluster, Ga is bonded in a 1-coordinate geometry to atoms. In the ScNO2 framework, Sc is bonded to two equivalent N, two equivalent O, and two equivalent O atoms to form corner-sharing ScN2O4 octahedra. The corner-sharing octahedral tilt angles range from 0-1°. Both Sc-N bond lengths are 2.09 Å. Both Sc-O bond lengths are 2.09 Å. Both Sc-O bond lengths are 2.09 Å. N is bonded in a linear geometry to two equivalent Sc atoms. There are two inequivalent O sites. In the first O site, O is bonded in a linear geometry to two equivalent Sc atoms. In the second O site, O is bonded in a linear geometry to two equivalent Sc atoms. The formation energy per atom is 1.4796. | Below is a description of a bulk material. The chemical formula is ScGaNO2. The elements are Sc, Ga, N, O. The formation energy per atom is positive. The spacegroup number is 24. The crystal system is orthorhombic. Generate the material: |  |  |
| OsAuO3 is (Cubic) Perovskite structured and crystallizes in the cubic Pm-3m space group. Os is bonded to six equivalent O atoms to form OsO6 octahedra that share corners with six equivalent OsO6 octahedra and faces with eight equivalent AuO12 cuboctahedra. The corner-sharing octahedra are not tilted. All Os-O bond lengths are 1.97 Å. Au is bonded to twelve equivalent O atoms to form distorted AuO12 cuboctahedra that share corners with twelve equivalent AuO12 cuboctahedra, faces with six equivalent AuO12 cuboctahedra, and faces with eight equivalent OsO6 octahedra. All Au-O bond lengths are 2.79 Å. O is bonded in a linear geometry to two equivalent Os and four equivalent Au atoms. The formation energy per atom is 1.4248. | Below is a description of a bulk material. The chemical formula is OsAuO3. The elements are Os, Au, O. The formation energy per atom is 1.4248. The space-group number is 220. The crystal system is cubic. Generate the material. |  |  |

Table 11: Visualization of the generated structures given textual description for **Perov-5** dataset

| Detailed Description | Short Prompt | Ground truth | Generated Samples |
|---|---|---|---|
| C crystallizes in the triclinic P1 space group. There are twenty-two inequivalent C sites. In the first C site, C(1) is bonded to one C(18), one C(5), and two equivalent C(9) atoms to form corner-sharing CC4 tetrahedra. …two equivalent C(12) atoms to form a mixture of distorted corner and edge-sharing CC4 trigonal pyramids. The energy per atom is -154.1336. | Below is a description of a bulk material. The chemical formula is C. The elements are C. The energy per atom is negative. The spacegroup number is 0. The crystal system is triclinic. Generate the material | | |
| C crystallizes in the orthorhombic Cmcm space group. There are two inequivalent C sites. In the first C site, C(1) is bonded to one C(2) and three equivalent C(1) atoms to form a mixture of corner and edge-sharing CC4 trigonal pyramids. The C(1)-C(2) bond length is 1.49 Å. There are two shorter (1.51 Å) and one longer (1.56 Å) C(1)-C(1) bond length. In the second C site, C(2) is bonded to one C(1) and three equivalent C(2) atoms to form corner-sharing CC4 tetrahedra. There are two shorter (1.54 Å) and one longer (1.56 Å) C(2)-C(2) bond length. The energy per atom is -154.2425. | Below is a description of a bulk material. The chemical formula is C. The elements are C. The energy per atom is negative. The spacegroup number is 62. The crystal system is orthorhombic. Generate the material. | | |
| C crystallizes in the triclinic P-1 space group. There are six inequivalent C sites. In the first C site, C(1) is bonded to one C(3), one C(5), and two equivalent C(4) atoms to form corner-sharing CC4 tetrahedra. …In the sixth C site, C(6) is bonded to one C(2), one C(4), and two equivalent C(3) atoms to form distorted corner-sharing CC4 tetrahedra. The energy per atom is -154.1338. | Below is a description of a bulk material. The chemical formula is C. The elements are C. The energy per atom is negative. The spacegroup number is 1. The crystal system is triclinic. Generate the material | | |
| C is a Theoretical Carbon Structure-like structure and crystallizes in the triclinic P-1 space group. There are nine inequivalent C sites. In the first C site, C(1) is bonded to one C(5), one C(6), one C(7), and one C(8) atom to form a mixture of corner and edge-sharing CC4 tetrahedra. The C(1)-C(5) bond length is 1.51 Å. The C(1)-C(6) bond length is 1.56 Å. The C(1)-C(7) bond length is 1.54 Å. …In the ninth C site, C(9) is bonded to one C(4), one C(6), one C(7), and one C(8) atom to form a mixture of corner and edge-sharing CC4 tetrahedra. The energy per atom is -154.2197. | Below is a description of a bulk material. The chemical formula is C. The elements are C. The energy per atom is -154.2197. The spacegroup number is 1. The crystal system is triclinic. Generate the material. | | |

Table 12: Visualization of the generated structures given textual description for **Carbon-24** dataset

| Detailed Description | Short Prompt | Ground truth | Generated Samples |
|---|---|---|---|
| Eu2PCl is Caswellsilverite-like structured and crystallizes in the tetragonal P4/mmm space group. There are two inequivalent Eu sites. In the first Eu site, Eu(1) is bonded to two equivalent P(1) and four equivalent Cl(1) atoms to form EuP2Cl4 octahedra that share corners with six equivalent Eu(1)P2Cl4 octahedra, edges with four equivalent Eu(1)P2Cl4 octahedra, and edges with eight equivalent Eu(2)P4Cl2 octahedra. ...The corner-sharing octahedra are not tilted. The formation energy per atom is -1.7615. The band gap is zero. The energy above the convex hull is zero. | Below is a description of a bulk material. The chemical formula is Eu2PCl. The elements are Eu, P, and Cl. The formation energy per atom is -1.7615. The band gap is 0.0. The energy above the convex hull is 0.0. The spacegroup number is 122. The crystal system is tetragonal. Generate the material. | | |
| MgNdHg2 is Heusler structured and crystallizes in the cubic Fm-3m space group. Mg(1) is bonded in a body-centered cubic geometry to eight equivalent Hg(1) atoms. All Mg(1)-Hg(1) bond lengths are 3.18 Å. Nd(1) is bonded in a body-centered cubic geometry to eight equivalent Hg(1) atoms. All Nd(1)-Hg(1) bond lengths are 3.18 Å. Hg(1) is bonded in a body-centered cubic geometry to four equivalent Mg(1) and four equivalent Nd(1) atoms. The formation energy per atom is -0.4708. The band gap is 0.0. The energy above the convex hull is 0.0. The spacegroup number is 224. | Below is a description of a bulk material. The chemical formula is NdMgHg2. The elements are Nd, Mg, and Hg. The formation energy per atom is -0.4708. The band gap is 0.0. The energy above the convex hull is 0.0. The spacegroup number is 224. The crystal system is cubic. Generate the material. | | |
| MgNdTl crystallizes in the hexagonal P-62m space group. Mg(1) is bonded in a 4-coordinate geometry to two equivalent Tl(1) and two equivalent Tl(2) atoms. Both Mg(1)-Tl(1) bond lengths are 3.01 Å. Both Mg(1)-Tl(2) bond lengths are 3.03 Å. Nd(1) is bonded in a 5-coordinate geometry to one Tl(2) and four equivalent Tl(1) atoms. The Nd(1)-Tl(2) bond length is 3.31 Å. All Nd(1)-Tl(1) bond lengths are 3.32 Å. There are two inequivalent Tl sites. In the first Tl site, Tl(2) is bonded in a distorted q6 geometry to six equivalent Mg(1) and three equivalent Nd(1) atoms. In the second Tl site, Tl(1) is bonded in a 9-coordinate geometry to three equivalent Mg(1) and six equivalent Nd(1) atoms. The formation energy per atom is -0.355. The band gap is 0.0. The energy above the convex hull is 0.0. The spacegroup number is 188. | Below is a description of a bulk material. The chemical formula is NdMgTl. The elements are Nd, Mg, and Tl. The formation energy per atom is -0.355. The band gap is 0.0. The energy above the convex hull is 0.0. The spacegroup number is 188. The crystal system is hexagonal. Generate the material. | | |
| LaNi2Ge2 crystallizes in the tetragonal I4/mmm space group. La(1) is bonded in a 16-coordinate geometry to eight equivalent Ni(1) and eight equivalent Ge(1) atoms. All La(1)-Ni(1) bond lengths are 3.25 Å. All La(1)-Ge(1) bond lengths are 3.26 Å. Ni(1) is bonded in a 4-coordinate geometry to four equivalent La(1) and four equivalent Ge(1) atoms. All Ni(1)-Ge(1) bond lengths are 2.39 Å. Ge(1) is bonded in a 9-coordinate geometry to four equivalent La(1), four equivalent Ni(1), and one Ge(1) atom. The Ge(1)-Ge(1) bond length is 2.66 Å. The formation energy per atom is -0.691. The band gap is 0.0. The energy above the convex hull is 0.0. | Below is a description of a bulk material. The chemical formula is La(NiGe)2. The elements are La, Ni, and Ge. The formation energy per atom is -0.691. The band gap is 0.0. The energy above the convex hull is 0.0. The spacegroup number is 138. The crystal system is tetragonal. Generate the material. | | |

Table 13: Visualization of the generated structures given textual description for **MP-20** dataset

