# OpenReview forum: "Periodic Materials Generation using Text-Guided Joint Diffusion Model"
_ICLR.cc/2025/Conference — ICLR 2025 Poster_

### Official Review · Reviewer_mJeo · 2024-10-26

**Soundness:** 2
**Presentation:** 3
**Contribution:** 2
**Rating:** 6
**Confidence:** 4

**Summary:**

This paper presents a text-guided diffusion model for periodic material generation. Compared to the baseline methods, the proposed text-guided approach achieves more valid and stable performance on benchmark datasets.

**Strengths:**

1. Integrating text information to guide crystal structure generation is a novel practice in the field of material structure generation.
2. From the experimental results, the method proposed in this paper achieves better performance compared to baseline methods. Additionally, the proposed method also demonstrates improved generation efficiency.

**Weaknesses:**

1. Although combining text information in the field of material generation is an innovative approach, the text-guided diffusion model is not a new model framework. Therefore, the contribution of this paper is limited from the perspective of architectural innovation.
2. TGDMat(Short) does not always outperform baseline methods, and the improvement of TGDMat(Long) over baseline methods is also marginal. This indicates that the model's results are heavily dependent on how the text information is constructed, and richer textual information can lead to slightly better outcomes than the baseline.
3. As metrics commonly used to evaluate generative performance, the authors should consider including diversity and novelty as additional metrics for comparison.

**Questions:**

See Weaknesses.

---

> ### Author Response · Authors · 2024-11-18
> **Point-by-point responses to Reviewer mJeo (1/2)**
>
> Dear Reviewer mJeo,
>
> Thank you very much for providing us with valuable feedback. We appreciate the detailed comments. Below, we have provided point-by-point responses to each of your comments.
>
> Weaknesses
> -
>
> - ***Regarding Architectural Innovation of TGDMat.***
> >
> >While text-guided diffusion models are widely used in fields such as image, video, and molecule generation, their application in periodic material generation remains largely unexplored. This study is the first to introduce text-guided diffusion for material generation. To achieve this, we have made the following novel contributions:
> >
> > - **Dataset Curation:** At the start of this work, no textual data was available for materials in benchmark databases such as Perov, Carbon, and MP. To address this, we curated textual data for these material databases, including both long and short textual prompts. Details of this curation process are provided in Section 4.2 and Appendix D: *Textual Dataset*. We plan to release these datasets to the community, anticipating that they will facilitate further exploration and research.
> > - **Simple Yet Effective Fusion Approach:** In TGDMat, we employ a straightforward yet highly effective method to integrate text-based contextual representations into the denoising network. Specifically, at each stage of denoising, we utilize the contextual information $\textbf{C}_\textbf{p}$ derived from textual descriptions via a pre-trained MatSciBERT model and concatenate it with the input atom features.
> >
> >
> >Our innovation, however, goes beyond text-guided diffusion. We further enhance the base diffusion model by introducing **Discrete Diffusion for Atom Types.** Prior works like CDVAE, DiffCSP used continuous diffusion on atom types (represented as the probability distribution for k classes and applying DDPM to learn the distribution). However, atom types are discrete data and it is well established in the literature[1] that using a continuous diffusion model for discrete features is not reasonable and produces suboptimal results. Hence we consider discrete diffusion on atom types in TGDMat, where we consider A as N discrete variables belonging to k classes and leverage the discrete diffusion model (D3PM)
> >
> >
> >[1] Andrew Campbell et.al.  A continuous time framework for discrete denoising models. Advances in Neural Information Processing Systems 35 (2022), 28266–28279.
>
> - ***Regarding Diversity and Novelty as additional metrics for comparison.***
> >
> >  - ***Regarding Novelty:*** For evaluation, we adopted the benchmark metrics introduced by CDVAE and subsequently used by prior works such as SyMat and DiffCSP. As outlined by Xie et al. in the CDVAE paper (Section 5.2: "Material Generation"), these seven metrics in the Generation Task aim to assess the novelty, validity, and property statistics of the generated materials.
> >
> > - ***Regarding Diversity:*** Since our paper focuses on conditional material generation, the goal is not to achieve diverse random generation but rather to produce constrained and targeted outputs that align with the criteria specified in the text descriptions of the test dataset. Consequently, the diversity of the generated materials depends on the diversity of the test dataset—if the test dataset is diverse, the generated materials will reflect that diversity; otherwise, they will not.

---

> ### Author Response · Authors · 2024-11-18
> **Point-by-point responses to Reviewer mJeo (2/2)**
>
> ***TGDMat(Short) does not always outperform baseline methods.***
> >
> >In Table 3, the text-guided baseline models such as CDVAE+, SyMat+, and DiffCSP+ are based on the long variant of their respective models. As a result, TGDMat(Short) does not consistently outperform these baseline methods, although it does outperform them in 7 out of 21 cases, ranking as the second-best model overall. To provide a more equitable comparison, we have separated the results of the short variants in the following table for both tasks.
> >
> >Gen Task (on Shorter Prompt) :
> > -
> >**Perov Dataset:**
> >
> >|Method|Comp Validity|Struct Validity|Cov-R|Cov-P|#Element|Density| Energy |
> >|-|-|-|-|-|-|-|-|
> >|CDVAE+(Short)  |98.17|100|99.40|99.01|0.0706|0.1395|0.0246|
> >|SyMat+(Short)  |96.94|100|99.22|98.40|0.0192|0.1827|0.2633|
> >|DiffCSP+(Short)|98.21|100|99.61|98.39|0.0123|0.1193|0.0266|
> >|TGDMat(Short)  |**98.28**|**100**|**99.71**|**99.24**|**0.0108**|**0.0947**|**0.0237**|
> >
> >**Carbon Dataset:**
> >
> >|Method|Comp Validity|Struct Validity|Cov-R|Cov-P|#Element|Density| Energy |
> >|-|-|-|-|-|-|-|-|
> >|CDVAE+(Short)  ||100|99.34|82.96||0.1398|0.2804|
> >|SyMat+(Short)  ||100|99.52|97.20||0.1206|3.7422|
> >|DiffCSP+(Short)||100|99.65|97.29||0.0811|0.0870|
> >|TGDMat(Short)  ||**100**|**99.81**|91.77||**0.0681**|**0.0865**|
> >
> >**MP Dataset:**
> >
> >|Method|Comp Validity|Struct Validity|Cov-R|Cov-P|#Element|Density| Energy |
> >|-|-|-|-|-|-|-|-|
> >|CDVAE+(Short)  |87.05|100|99.36|99.60|0.9930|0.6420|0.2970|
> >|SyMat+(Short)  |88.08|99.9|98.59|99.47|0.5031|0.3917|0.3622|
> >|DiffCSP+(Short)|84.57|100|99.52|99.85|0.3310|0.3800|0.1379|
> >|TGDMat(Short)  |86.60|**100**|**99.79**|**99.88**|0.3337|**0.3296**|**0.1189**|
> >
> > We observe that among the total 19 metrics over three datasets in the table, TGDMat(Short) method outperforms all other baselines in 16 metrics.
> >
> >
> >CSP Task (on Shorter Prompt) :
> > -
> >|Method|#Samples|Perov|Perov|Carbon|Carbon|MP|MP|
> >|-|-|-|-|-|-|-|-|
> >|||Match Rate|RMSE|Match Rate|RMSE|Match Rate|RMSE|
> >|CDVAE+(Short)|1 |48.97|0.1063|22.65|0.2640|40.33|0.1037|
> >|CDVAE+(Short)|20|89.54|0.0423|89.61|0.2188|70.22|0.0876|
> >|SyMat+(Short)|1 |49.39|0.0985|23.71|0.2567|40.84|0.1027|
> >|SyMat+(Short)|20|92.10|0.0255|90.86|0.2069|71.31|0.0875|
> >|DiffCSP+(Short) or TGDMat (Short)                           |1|**56.54**|**0.0583**|**24.13**|**0.2424**|**52.22**|**0.0597**|
> >|DiffCSP+(Short) or TGDMat (Short) |20|**98.25**|**0.0137**|**88.28**|**0.2252**|**80.97**|**0.0443**|
> >
> >In CSP Task, we observe TGDMat (Short) performs better than all the baselines across dataset in both setup: using number of samples 1 and 20. Please note in the CSP task, since atom types are given DiffCSP+ (Short) and TGDMat (Short) are the same architecture.
>
> ***Marginal Improvement of TGDMat(Long) over baseline methods.***
> >
> >We respectfully disagree with the reviewer’s assessment that TGDMat(Long) offers only marginal improvements.
> >
> >Gen Task (on Long Prompt) :
> > -
> > For metrics where baselines already achieve over 98% accuracy (e.g., Structural and Compositional Validity for Perov, Carbon, or Coverage Metrics), the scope for further improvement is inherently limited. However, for other metrics, TGDMat(Long) demonstrates significant enhancements. Specifically, the average improvement in property statistics are as follows:
> >
> >|  | Perov | Carbon | MP |
> >| -------- | -------- | -------- |-------- |
> >| Avg Improvement by TGDMat(Long)   | 35.61%     | 38.91%     |14.57%
> >
> >Moreover, for the **MP-20 dataset, compositional validity is improved by 5%, which is a notable advancement.**
> >Overall, TGDMat’s superior performance across seven metrics on the MP-20 dataset underscores its potential to generate novel materials that are experimentally synthesizable.
> >
> >CSP Task (on Long Prompt) :
> > -
> > Prior unconditional diffusion models demonstrate improved Match Rates and lower RMSE when generating 20 samples (k = 20) per test material, they largely fail in both metrics when generating only one sample (k = 1) per test material. Using text guidance(Long Variant) during the reverse denoising process, with just one generated sample per test material, text-guided variants outperform respective vanilla models, thereby reducing computational overhead.
> >
> >Overall, these findings highlight the impact of our key contributions in TGDMat: 1) Joint Diffusion on A, X, and L, 2) Discrete Diffusion on A, and 3) Text-Guided Diffusion.
>
> Finally, we would like to thank Reviewer mJeo once again for these valuable comments. We will reflect these comments in the revised manuscript. We believe that our responses above address all of Reviewer mJeo's concerns and contribute to further strengthening our work.
>
> Sincerely,
> The Authors

---

> > ### Comment · Reviewer_mJeo · 2024-11-21
> >
> > Thank you for your detailed reply. I increased the rating score.

---

> > > ### Author Response · Authors · 2024-11-27
> > >
> > > Dear Reviewer mJeo,
> > >
> > > We sincerely thank you for your positive feedback and for raising the score to 6. We have carefully incorporated all your suggested changes into our revised manuscript, and we believe these contributions significantly enhance the research at the intersection of AI and material science.
> > >
> > > We would be grateful if you could review the revised version and let us know if it is possible to change the decision from borderline accept to accept.
> > >
> > >  Your feedback would be highly appreciated.    If you feel that additional experiments or results would further strengthen the manuscript, we are more than willing to provide them and actively engage in any further discussions.
> > >
> > > Thank you once again for your valuable insights and support. We look forward to your response.
> > >
> > > Thank you,
> > >
> > > The Authors

---

### Official Review · Reviewer_ijYE · 2024-10-30

**Soundness:** 3
**Presentation:** 3
**Contribution:** 3
**Rating:** 6
**Confidence:** 4

**Summary:**

The paper focuses on a text-guided diffusion model for periodic material generation. It first designs a diffusion model that can jointly model atom types, coordinates, and lattice structures for periodic materials, then proposes incorporating material structures and properties through text to enhance generation performance. In the experiments, it demonstrates improvements by incorporating text into existing diffusion models, such as SyMat and DiffCSP, followed by showing the superiority of the proposed TGDMat over existing baselines.

**Strengths:**

- This paper proposes the joint learning fashion for three types of crystal structural information. Text guided denoising network can be the right way to the solution of the problem.
- The paper is well-organized, self-contained with a comprehensive and detailed reference section.
- The design of textual description in the reverse diffusion process is convincing according to the supplementary material.
- The paper provides the computational advantage of integrating text knowledge during reverse diffusion compared to other baseline models.

**Weaknesses:**

- Motivation should be clarified: (a) Providing guidance in diffusion models for improved generation is important, but it is unclear why text must be used to incorporate such guidance. The guidance information in line 70 could also be modeled as feature vectors, reducing the need for text embedding models, which may not always provide accurate text-based embeddings. Additionally, text-based models may struggle to capture subtle numerical differences. Is there any prior work on using feature vectors for guidance? What are the advantages of using text rather than directly using numerical values or feature vectors? (b) The idea of studying joint diffusion over all three components of crystal structures: the lattices, atom types, and coordinates, by employing a periodic-E(3)-equivariant denoising model, kind of the same as the DiffCSP. Though exploring both crystal structure prediction (CSP) and random generation task (Gen), the text guided denoising network seems to be straightforward by further considering the CSP model architecture. It is not clear how much of this part of work is based on previous research and what is new.

- Method design should have rationales: The authors applied the diffusion process independently to three variables: lattice (line 210), atom types (line 248), and atom coordinates (line 272). However, in the reverse process, the denoising models use all variables with conditions for noise estimation (line 289). Given that lattice noise is independent of atom types/coordinates and vice versa, is there a reason for not simply using the lattice variable (plus $C_p$) for lattice denoising?

- Experiments should be improved: The paper lacks an evaluation of whether the generated materials meet the text descriptions. Coverage may reflect this to some extent, but it is too general and lacks details on specific properties mentioned in the descriptions. Property statistics are at the distribution level between two sets and lack point-to-point comparisons. Also, the paper lacks the ablation study for the denoising model to demonstrate the individual importance of each component (or the authors could highlight the part if I missed it).

**Questions:**

- In line 48, to address issues in local message passing, how about using Transformers?
- I am not sure if I missed anything, Figure 2 shows ‘SE(3)-Equivariant GNN model’.. Also, are the diffusion and reverse processes in Figure 2 are correct?
- The paper conducts joint diffusion on lattices, atom types, and coordinates, is there any ablation study for the separately and jointly learning of crystal geometry?
- The CDVAE employs SE(3) equivariant GNNs adapted with periodicity to ensure the invariance of materials. Why not use it and what’s the difference between SE(3) and periodic E(3)-equivariant denoising model backbone?
- Noticing there would be several generated samples in Table 4, how would one select the best sample if the ground truth was not available?

---

> ### Author Response · Authors · 2024-11-18
> **Point-by-point responses to Reviewer ijYE (1/3)**
>
> Dear Reviewer ijYE,
>
> Thank you very much for providing us with valuable feedback. We appreciate the detailed comments. Below, we have provided point-by-point responses to each of your comments.
>
> Weaknesses:
> -
>
> ***Utility of Text-guidace than Feature Vectors-guidance***
> >In literature [1], it has been extensively studied that using value guidance specifying a single or a handful of target properties or features as feature vectors might be insufficient to capture intricate conditions.  In contrast, textual descriptions allow us to encompass these conditions adeptly and flexibly and produce better results than value-based guidance. Also, Since we use MatSciBert which is pre-trained on a huge corpus of articles from materials domains, it allows us to encode the more enriched and robust contextual information about the tokens.
> >
> >[1] Luo, Yanchen, et al. "Text-guided diffusion model for 3d molecule generation." (2024).
> >
> > However, in response to the reviewer's question, we conducted an additional experiment, where we feed all relevant conditional information  e.g Formula, Space Group, Crystal Symmetry,Bond Length and Property Values as feature vectors to guide the diffusion model. Here are the results we observed for Gen Task:
> >
> > ***Perov Dataset:***
> >
> >|Method|Comp Validity|Struct Validity|Cov-R|Cov-P|#Element|Density| Energy |
> >|-|-|-|-|-|-|-|-|
> >|TGDMat (Conditions as Features)          |96.55|98.73|99.18|97.06|0.0149|0.1200|0.0290|
> >|TGDMat (Conditions as Text Emb)  |**98.63**|**100**|**99.87**|**99.52**|**0.0090**|**0.049**|**0.018**|
> >
> >***Carbon Dataset:***
> >
> >|Method|Comp Validity|Struct Validity|Cov-R|Cov-P|#Element|Density| Energy |
> >|-|-|-|-|-|-|-|-|
> >|TGDMat (Conditions as Features)  ||99.56 |99.32|92.17||0.104|0.087|
> >|TGDMat (Conditions as Text Emb)  ||**100** |**99.91**|**92.43**||**0.043**|**0.063**|
> >
> >***MP Dataset:***
> >
> >|Method|Comp Validity|Struct Validity|Cov-R|Cov-P|#Element|Density| Energy |
> >|-|-|-|-|-|-|-|-|
> >|TGDMat (Conditions as Features)          |83.85|99.61 |98.97|98.3|0.377|0.375|0.126|
> >|TGDMat (Conditions as Text Emb)          |**92.97**|**100** |**99.89**|**99.95**|**0.289**|**0.308**|**0.115**|
> >
> >Across three datasets, we did not observe performance improvements. We will add these results in the revised manuscripts.
>
> ***Key differences between DiffCSP and TGDMat***
> > Following are the key differences between DiffCSP and our proposed TGDMat
> >
> >|           | DiffCSP          | TGDMat   |
> >| --------  | --------         | -------- |
> >| Tasks     | Only CSP Task    | Both CSP and Gen Tasks|
> >| Diffusion on Atom Type | NA    | Discrete Diffusion (D3PM)|
> >| Model Category | Unconditional; unable to specify the criteria required by the user.    | Conditional; able to specify the criteria required by the user.(in Text Format)|
> >| Text Guided Diffusion     | NO    | Yes|
> >
> >However, note the goal of this paper is not to introduce a new diffusion model to replace existing models like DiffCSP or CDVAE for periodic material generation. Instead, **we focus on demonstrating that conditional models can outperform traditional unconditional models, such as DiffCSP**. Specifically, we show that incorporating textual conditions through text-guided diffusion leads to better performance compared to using unconditional models like DiffCSP. Additionally, we enhance DiffCSP by integrating discrete diffusion over atom types in our proposed TGDMat framework.
>
> ***Regarding Diffusion Method Design***
> > In the diffusion model, the forward process is non-parametric, meaning we simply add noise to three variables (atom coordinates, atom types, and the lattice) independently, without any learning involved. However, during the denoising process, a backbone Equivariant Graph Neural Network (EGNN) is used to predict the noise at each time step. At any given time t, the EGNN takes the atom types A_t, atom coordinates X_t, and lattice L_t together as input. Then the EGNN performs message passing and aggregation to generate node and graph representations, which are then used to predict the noise. Thus, the denoising process depends on all three variables.

---

> ### Author Response · Authors · 2024-11-18
> **Point-by-point responses to Reviewer ijYE (2/3)**
>
> ***Evaluation of whether the generated materials meet the text descriptions.***
> > Quantitative results on the alignment of the generated structures with the given prompts, in terms of compositions, properties, and other factors, are provided in the "**F.3 CORRECTNESS OF GENERATED MATERIALS**" section of the Appendix (Page 23, Line 1229). This is also referenced in the "Additional Results" section (Page 10, Line 527) of the main manuscript. Following are the details :
> >
> >
> > To ensure the fidelity of our model’s outputs concerning these specified global attributes from the text prompt, we randomly generated 1000 materials (sampled from all three Datasets) based on their respective textual descriptions(both Long and Short) and assessed the percentage of generated materials that matched the global features outlined in the text prompt. In specific, we matched the Formula, Space group, Crystal System, and Dimensions of generated materials with the textual descriptions. Moreover, we examined whether properties such as formation energy and bandgap matched the specified criteria as per the text prompt (positive/negative, zero/nonzero).
> >
> >
> > Results for TGDMat(Long):
> >
> > | Global Feature    |% of Matched Materials(Perov)|% of Matched Materials (Carbon)|% of Matched Materials(MP)|
> > | --------|--   |---- |--   |
> > | Formula           |97.50|98.20|70.54||
> > | Space Group       |87.00|80.79|67.88|
> > | Crystal System    |92.60|91.55|73.54|
> > | Formation Energy  |95.49|  -  |92.88|
> > | Band Gap          |  -  |98.61| 96.73|
> >
> > Results for TGDMat(Short):
> >
> > |Global Feature|% of Matched Materials(Perov)|% of Matched Materials (Carbon)|% of Matched Materials(MP)|
> > |-|-|-|-|
> > |Formula|90.70|92.56|65.22||
> > |Space Group|86.51| 80.50| 58.77|
> > |Crystal System|83.19| 81.64|72.77|
> > |Formation Energy|90.33|-| 91.00|
> > |Band Gap|-|95.90| 93.33|
> >
> > In general, using longer text, considering Perov-5 and Carbon-24 datasets, the generated material meets the specified criteria effectively. However, when dealing with the MP-20 dataset, which is more intricate due to its complex structure and composition, performance tends to decline. Additionally, when using shorter prompts, overall performance suffers across all datasets compared to longer text inputs. This is because the longer text, provided by the robocrystallographer, offers a comprehensive range of information, both global and local, thereby enhancing the generation capabilities of TGDMat.
>
> Questions
> -
>
> ***To address issues in local message passing, how about using Transformers?***
> >Replacing GNNs with Transformers as the backbone network is an exciting research direction, but implementing Transformers comes up with its own challenges[1]. One of the key obstacles is the scalability and efficiency of graph transformers, which require significant memory and computational resources, particularly when using global attention mechanisms. These challenges become even more pronounced in deeper architectures, which are more prone to overfitting and over-smoothing. Additionally, graph transformers often face challenges when it comes to generalizing to graphs. As a result, in many popular graph machine learning applications, graph transformers have not yet fully replaced the Message Passing (GNN) framework. We consider this an important area for future exploration.
> >
> >
> >However, in response to the reviewer's suggestion we conduct an additional study, where we replace the backbone of TGDMat with Matformer[2], a popular transformer model for crystal property prediction. We present the results of Gen Tasks in the following table and compare them with TDGMat:
> >
> >
> >**Perov Dataset:**
> >
> >
> >|Method|Comp Validity|Struct Validity|
> >|-|-|-|
> >|TGDMat (Matformer Backbone)|93.77|90.13
> >|TGDMat (GNN Backbone)|**98.63**|**100**
> >
> >**Carbon Dataset:**
> >
> >
> >|Method|Comp Validity|Struct Validity|
> >|-|-|-|
> >|TGDMat(Matformer Backbone)|-|89.26
> >|TGDMat(GNN Backbone)|-|**100**
> >
> >**MP Dataset:**
> >
> >
> >|Method|Comp Validity|Struct Validity|
> >|-|-|-|
> >|TGDMat(Matformer Backbone)|81.96|84.37
> >|TGDMat(GNN Backbone)|**92.97**|**100**
> >
> >We observe significant performance degradation in all metrics across datasets. This needs further exploration and we keep this a scope of future work.
> >
> >[1] Shehzad, A. Graph transformers: A survey. arXiv preprint arXiv:2407.09777.
> >
> >[2] Yan, Keqiang, et al." Periodic graph transformers for crystal material property prediction. NeuRIPs 35 (2022): 15066-15080.
>
> ***Regarding Typo in Fig-2.***
>
> > We apologize for the mistake and appreciate you pointing it out. The direction of the arrows for the diffusion and reverse processes was incorrect. For the diffusion/forward process, the arrow should point from $M_0$ to $M_T$, while for the denoising/reverse process, it should point from $M_T$ to $M_0$. Also, the GNN model would be a periodic E(3)-equivariant GNN model. We will correct this error and update the revised manuscript accordingly.

---

> ### Author Response · Authors · 2024-11-18
> **Point-by-point responses to Reviewer ijYE (3/3)**
>
> ***Ablation study for the separately and jointly learning of crystal geometry.***
> >In response to the reviewer's feedback we conducted an ablation study where we use three diffusion models to learn A,X,L separately. While sampling we sample A,X,L separately and merge them together. We fuse the textual representation in the same way in all three diffusion models. We present the results in following table and compare with TDGMat:
> >
> >***Perov Dataset:***
> >
> >
> >|Method|Comp Validity|Struct Validity|Cov-R|Cov-P|#Element|Density| Energy |
> >|-|-|-|-|-|-|-|-|
> >|TGDMat (Learning A,X,L seperately)          |90.10|85.43|85.77|83.51|0.341|0.591|0.376|
> >|TGDMat (Jontly Learning A,X,L) |**98.63**|**100**|**99.87**|**99.52**|**0.0090**|**0.049**|**0.018**|
> >
> >***Carbon Dataset:***
> >
> >
> >|Method|Comp Validity|Struct Validity|Cov-R|Cov-P|#Element|Density| Energy |
> >|-|-|-|-|-|-|-|-|
> >|TGDMat (Learning A,X,L seperately)             ||75.64 |80.95|82.29||0.435|0.584|
> >|TGDMat (Jontly Learning A,X,L)                ||**100** |**99.91**|**92.43**||**0.043**|**0.063**|
> >
> >***MP Dataset:***
> >
> >
> >|Method|Comp Validity|Struct Validity|Cov-R|Cov-P|#Element|Density| Energy |
> >|-|-|-|-|-|-|-|-|
> >|TGDMat (Learning A,X,L seperately)          |73.18|77.01|82.99|72.41|0.861|0.887|0.634|
> >|TGDMat (Jontly Learning A,X,L)           |**92.97**|**100** |**99.89**|**99.95**|**0.289**|**0.308**|**0.115**|
> >
> >We observe significant performance degradation in all metrics across datasets if we learn A,X,L separately. We will add these results in the revised manuscripts.
>
> ***What’s the difference between SE(3) and periodic E(3)-equivariant denoising model backbone?***
> >A model is An SE(3) equivariant if it follows invariance to permutation, translation, and rotation. Additionally, for crystals, we need invariance for periodic transformations, since the atoms in the unit cell can periodically repeat themselves infinite times along the lattice vector, there can be many choices of unit cells and coordinate matrices representing the same material. We denote this as a periodic E(3)-equivariant denoising model or GNN. SE(3) equivariant GNNs adapted with periodicity proposed by CDVAE are also similar to the GNN model that follows invariance to permutation, translation, rotation, and periodicity. More details about the physical symmetry of the crystal structure are provided in ***Appendix C: "INVARIANCES IN CRYSTAL STRUCTURE."***
>
> ***How would one select the best sample if the ground truth was not available?***
> > In the absence of ground truth, selecting the best sampleamong generated materials involves ensuring things such as the generated samples adhere to fundamental physical and chemical principles, such as proper bonding patterns, reasonable interatomic distances, and compliance with crystal symmetry constraints. One can also use tools like DFT to calculate formation energy, energy above the convex hull, or phonon stability to select the best samples. However, at this point of time there is no single agreed method to choose the best sample.
>
> Finally, we would like to thank Reviewer ijYE once again for these valuable comments. We will reflect these comments in the revised manuscript. We believe that our responses above address all of Reviewer ijYE's concerns and contribute to further strengthening our work.
>
> Sincerely,
> The Authors

---

> ### Comment · Reviewer_ijYE · 2024-11-21
> **Response**
>
> I thank the authors for delivering additional results and discussions quickly to answer my questions. Incorporating the results and discussions can improve the soundness of the work. I increase the soundness score from 2 to 3. My evaluation on the contribution remains. I may put the overall score between 6 and 7, however, based on the assessment on novelty and contribution, it cannot reach 8 in my evaluation. So I'd keep the overall score the same.

---

> > ### Author Response · Authors · 2024-11-22
> > **Response to Reviewer ijYE's Feedback**
> >
> > We sincerely thank the reviewer for their positive feedback and for increasing the soundness score to 3. We also appreciate the reviewer’s acknowledgment that the overall score of the paper should exceed 6 (above the borderline).
> >
> > Below, we summarize the novelty and contributions of our work:
> >
> > - Exploring Text-Guided Diffusion for Periodic Material Generation
> > - Curating Text Datasets for Benchmark Databases
> > - A Simple Yet Effective Fusion Approach to Integrate Text into the Denoising Process
> > - Introducing Discrete Diffusion for Atom Types
> > - Joint Diffusion on A, X, and L
> >
> > We believe these contributions play a meaningful role in advancing research at the intersection of AI and material science.
> > If the reviewer requires **additional experiments or results to improve the work further**, we are happy to provide them and actively participate in the discussion.
> >
> > As suggested, we will incorporate the results and discussions into the revised manuscript.

---

> > > ### Author Response · Authors · 2024-11-25
> > > **Eagerly Waiting for further feedback**
> > >
> > > Dear Reviewer,
> > >
> > > Thank you for your valuable feedback and thoughtful comments.
> > >
> > > In response, we have uploaded the revised manuscript and made the following updates as per your suggestions:
> > >
> > >
> > > - Assessing how well the generated materials align with the provided text prompts (Section 5.4)
> > > - Comparing the utility of text-guidance versus feature vector-guidance (Appendix F.7)
> > > - Ablation study on the joint learning of crystal geometry (Appendix F.8)
> > > - Providing a detailed discussion on the key differences between DiffCSP and TGDMat.(Section 3 and Appendix B.5)
> > > - Correcting a typo in Figure 2 regarding the direction of forward and reverse diffusions.
> > >
> > >
> > > We hope these revisions adequately address all the concerns you raised and improved the quality of our manuscript. If there are any remaining issues or additional clarifications needed, please let us know, and we would be happy to address them. Otherwise, we kindly request you to consider revising the score based on these updates.
> > >
> > > We look forward to your response.
> > >
> > > Thank you,
> > >
> > > The Authors

---

### Official Review · Reviewer_D75u · 2024-11-03

**Soundness:** 3
**Presentation:** 2
**Contribution:** 2
**Rating:** 6
**Confidence:** 3

**Summary:**

The paper introduces a text-guided diffusion model for generating 3D periodic materials. The work leverages a periodic-E(3)-equivariant graph neural network (GNN) to jointly generate atom types, fractional coordinates, and lattice structures, while integrating textual descriptions at each denoising step.

**Strengths:**

1. The model effectively learns the joint distribution of atom coordinates, types, and lattice structure through an end-to-end diffusion network, which is a significant improvement over existing models that handle these aspects separately.
2. Incorporating textual descriptions as a condition during the denoising process enhances the model's ability to generate materials that meet specific user-defined criteria, making it more versatile and user-friendly.

**Weaknesses:**

1. The paper does not provide sufficient detail on how the contextual representation of long, detailed text data is fused into the denoising network to generate text-guided variants of baseline models.
2. The visualization of de novo generated materials lacks a thorough discussion on whether the results align with the given prompts. Additionally, the alignment of generated materials with general textual conditions is not adequately discussed, which is crucial for validating the model's performance.
3. There is a need for a more comprehensive ablation study comparing the effects of long, detailed descriptions versus short prompts on the model's guidance performance. This would help understand the robustness and versatility of the model under different input conditions.

**Questions:**

1. Could the authors provide more details on how the contextual representation of long, detailed text data is integrated into the denoising network? What specific techniques or architectures are used to achieve this?
2. Could the authors include an ablation study comparing long and short text prompts on the baselines in Table 1 or 2? How would the results differ, and what insights could be gained from such a study?
3. Can the authors provide more detailed visualizations of the generated materials and explicitly discuss how well these results align with the given text prompts? Are there any metrics or qualitative assessments to evaluate this alignment?

---

> ### Author Response · Authors · 2024-11-18
> **Point-by-point responses to Reviewer D75u (1/2)**
>
> Dear Reviewer D75u,
>
> Thank you very much for providing us with valuable feedback. We appreciate the detailed comments. Below, we have provided point-by-point responses to each of your comments.
>
>
> Questions:
> -
> ***Details on how the contextual representation of long, detailed text data is integrated into the denoising network?***
> >
> > As mentioned in ***Section 4.3.2 TEXT GUIDED DENOISING NETWORK (Line-313-18), Equation-5***, At each timestep t of reverse diffusion, we concatenate textual representation $\textbf{C}_\textbf{p}$ with each input atom feature. Following this same approach, we also developed text-guided versions of the baseline models, named CDVAE+, SyMat+, and DiffCSP+, in which the contextual representation from detailed text data is integrated into the denoising networks of these models.
>
> ***How well generated materials align with the given text prompts?***
> >
> > Quantitative results on the alignment of the generated structures with the given prompts, in terms of compositions, properties, and other factors, are provided in the "***F.3 CORRECTNESS OF GENERATED MATERIALS***" section of the Appendix (Page 23, Line 1229). This is also referenced in the "Additional Results" section (Page 10, Line 527) of the main manuscript.  ***In the revised manuscript, this has been incorporated into Section 5.4 of the main manuscript.***
> >
> >Following are the details :
> >
> > To ensure the fidelity of our model’s outputs concerning these specified global attributes from the text prompt, we randomly generated 1000 materials (sampled from all three Datasets) based on their respective textual descriptions(both Long and Short) and assessed the percentage of generated materials that matched the global features outlined in the text prompt. In specific, we matched the Formula, Space group, and Crystal System, and Dimensions of generated materials with the textual descriptions.Moreover, we examined whether properties such as formation energy and bandgap matched the specified criteria as per the text prompt (positive/negative, zero/nonzero).
> >
> >
> > Results for TGDMat(Long):
> >
> > | Global Feature    |% of Matched Materials(Perov)|% of Matched Materials (Carbon)|% of Matched Materials(MP)|
> > | --------          |--   |---- |--   |
> > | Formula           |97.50|98.20|70.54||
> > | Space Group       |87.00|80.79|67.88|
> > | Crystal System    |92.60|91.55|73.54|
> > | Formation Energy  |95.49|  -  |92.88|
> > | Band Gap          |  -  |98.61| 96.73|
> >
> > Results for TGDMat(Short):
> >
> > | Global Feature    |% of Matched Materials(Perov)|% of Matched Materials (Carbon)|% of Matched Materials(MP)|
> > | --------          |--   |---- |--   |
> > | Formula           |90.70|92.56|65.22||
> > | Space Group       |86.51| 80.50| 58.77|
> > | Crystal System    |83.19| 81.64|72.77|
> > | Formation Energy  |90.33|  -  | 91.00|
> > | Band Gap          |  -  |95.90| 93.33|
> >
> > In general, using longer text, considering Perov-5 and Carbon-24 datasets, the generated material meets the specified criteria effectively. However, when dealing with the MP-20 dataset, which is more intricate due to its complex structure and composition, performance tends to decline. Additionally, when using shorter prompts, overall performance suffers across all datasets compared to longer text inputs. This is because the longer text, provided by the robocrystallographer, offers a comprehensive range of information, both global and local, thereby enhancing the generation capabilities of TGDMat.

---

> ### Author Response · Authors · 2024-11-18
> **Point-by-point responses to Reviewer D75u (2/2)**
>
> ***Ablation study comparing long and short text prompts on the baseline models.***
> >
> > We begin by presenting the ablation study results comparing long and short text prompts on both tasks, followed by key insights and observations.
> >
> > Gen Task:
> > -
> >
> > **Perov**
> >
> >|Method|Comp Validity|Struct Validity|Cov-R|Cov-P|#Element|Density| Energy |
> >|-|-|-|-|-|-|-|-|
> >|CDVAE|98.29|100|99.25|98.39|0.0731|0.1462|0.0291|
> >|CDVAE+(Short)|98.37|100|99.40|99.01|0.0706|0.1395|0.0246|
> >|CDVAE+(Long)|98.45|100|99.53|99.09|0.0609|0.1276|0.0223|
> >|SyMat|96.83|100|99.16|98.29|0.0193|0.1991|0.2827|
> >|SyMat+(Short)|96.94|100|99.22|98.40|0.0192|0.1827|0.2633|
> >|SyMat+(Long)|97.88|100|99.71|98.79|0.0172|0.1755|0.2566|
> >|DiffCSP|98.15|100|99.28|98.08|0.0132|0.1281|0.0267|
> >|DiffCSP+(Short)|98.21|100|99.61|98.39|0.0123|0.1193|0.0266|
> >|DiffCSP+(Long) |98.44|100|99.85|98.53|0.0119|0.1071|0.0241|
> >|TGDMat(Short)|98.28|100|99.71|99.24|0.0108|0.094|0.023|
> >|TGDMat(Long)|**98.63**|**100**|**99.87**|**99.52**|**0.009**|**0.049**|**0.018**|
> >
> > **Carbon**
> >
> >|Method|Comp Validity|Struct Validity|Cov-R|Cov-P|#Element|Density| Energy |
> >|-|-|-|-|-|-|-|-|
> >|CDVAE||100 |99.35|82.66||0.1539|0.2889|
> >|CDVAE+(Short)||100|99.34|82.96||0.1398|0.2804|
> >|CDVAE+(Long)||100|99.82|84.76||0.1377|0.2660|
> >|SyMat||100 |99.42|97.17||0.1234|3.9628|
> >|SyMat+(Short)||100|99.52|97.20||0.1206|3.7422|
> >|SyMat+(Long)||100|99.90|**97.63**||0.1171|3.8620|
> >|DiffCSP||99.9|99.49|97.27||0.0861|0.0876|
> >|DiffCSP+(Short)||100|99.61|97.29||0.0811|0.087 |
> >|DiffCSP+(Long)||100|**99.93**|97.33||0.0763|0.0853|
> >|TGDMat(Short)||100|99.81|91.77||0.0681|0.0865|
> >|TGDMat(Long)||**100**|99.9|92.43||**0.043**|**0.063**|
> >
> >**MP**
> >
> >|Method|Comp Validity|Struct Validity|Cov-R|Cov-P|#Element|Density| Energy |
> >|-|-|-|-|-|-|-|-|
> >|CDVAE|86.30|100 |99.15|99.49|1.4921|0.7085|0.3039|
> >|CDVAE+(Short)|87.05|100 |99.36|99.60|0.9930|0.6420|0.2970|
> >|CDVAE+(Long)|87.42|100 |99.57|99.81|0.9720|0.6388|0.2977|
> >|SyMat|87.96|99.9|98.30|99.37|0.5236|0.4012|0.3877|
> >|SyMat+(Short)|96.94|99.9|98.59|99.47|0.5031|0.3917|0.3622|
> >|SyMat+(Long)|97.88|99.9|99.01|99.95|0.4865|0.3879|0.3489|
> >|DiffCSP|83.25|100 |99.41|99.76|0.3411|0.3802|0.1497|
> >|DiffCSP+(Short)|84.57|100 |99.52|99.85|0.3310|0.3800|0.1379|
> >|DiffCSP+(Long)|85.07|100 |99.81|99.89|0.3122|0.3799|0.1355|
> >|TGDMat(Short)|86.60|100 |99.79|99.88|0.3337|0.3296|0.1189|
> >|TGDMat(Long)|**92.97**|**100**|**99.89**|**99.95**|**0.289**|**0.308**|**0.115**|
> >
> > CSP Task :
> > -
> >
> >|Method|#Samples|Perov|Perov|Carbon|Carbon|MP|MP|
> >|-|-|-|-|-|-|-|-|
> >|||Match Rate|RMSE|Match Rate|RMSE|Match Rate|RMSE|
> >|CDVAE|1|45.31|0.1138|17.09|0.2969|33.90|0.1045|
> >|CDVAE|20|88.51|0.0464|88.37|0.2286|66.95|0.1026|
> >|CDVAE+(Short)|1 |48.97|0.1063|22.65|0.2640|40.33|0.1037|
> >|CDVAE+(Short)|20|89.54|0.0423|89.61|0.2188|70.22|0.0876|
> >|CDVAE+(Long)|1 |49.25|0.1055|23.73|0.2590|41.80|0.1021|
> >|CDVAE+(Long)|20|89.73|0.0417|89.77|0.2053|72.56|0.0840|
> >|SyMat|1|47.32|0.1074|20.81|0.2655|33.92|0.1039|
> >|SyMat|20|90.25|0.0316|89.29|0.2184|71.03|0.0945|
> >|SyMat+(Short)|1 |49.39|0.0985|27.71|0.2567|40.84|0.1027|
> >|SyMat+(Short)|20|92.10|0.0255|90.86|0.2069|71.31|0.0875|
> >|SyMat+(Long)|1 |50.88|0.0963|28.18|0.2510|43.17|0.1016|
> >|SyMat+(Long)|20|92.30|0.0201|91.65|0.1870|72.96|0.0820|
> >|DiffCSP|1|52.02|0.0760|17.54|0.2759|51.49|0.0631|
> >|DiffCSP|20|98.60|0.0128|88.47|0.2192|77.93|0.0492|
> >|DiffCSP+(Short)|1 |56.54|0.0583|24.13|0.2424|52.22|0.0597|
> >|DiffCSP+(Short)|20|98.25|0.0137|88.28|0.2252|80.97|0.0443|
> >|DiffCSP+(Long)|1 |90.46|0.020|44.63|0.226|55.15|0.057|
> >|DiffCSP+(Long)|20|**98.59**|**0.007**|**95.27**|**0.153**|**82.02**|**0.039**|
> >
> > Observations:
> > 1. For both tasks, across all the datasets, text guidance out performs the vanilla diffusion models in almost all metrics.
> > 2. Our experiments suggest that using shorter prompts textguided models outperforms the vanilla baseline models. However performance is even superior when using text guided diffusions using longer prompts.
> > 3. For CSP task, using text guidance during the reverse denoising process, with just one generated sample per test material, text-guided variants outperform respective vanilla models, thereby reducing computational overhead.
> > 4. Our proposed TGDMat (Long) stands out as the leading model when compared to all baseline models and their text-guided variants across three benchmark datasets. In specific, for Gen Task, TGDMat (Long) outperforms closest baseline DiffCSP+ (Long) because we leveraged discrete diffusion on atom types, which is more powerful learning discrete variables like atom types.
> > 5. Finally, results indicate that utilizing shorter prompts TGDMat (Short) results in a slight decrease in overall performance compared to the longer variant TGDMat (Long). Nonetheless, the performance remains superior or comparable to baseline models (vanilla and text-guided variants)
> >
> >We will add these comprehensive results in the revised manuscripts ***(Appendix F.3 in the revised version)***.

---

> ### Author Response · Authors · 2024-11-23
> **Looking forward to your feedback**
>
> Dear Reviewer,
>
> Thank you for your valuable feedback and constructive comments.
>
> In our rebuttal, we have provided additional experiments and enhanced explanations, and we hope we have addressed all the concerns raised by the reviewer. We are open to further discussions and are happy to clarify any remaining doubts.
>
> If there are still any outstanding issues, we kindly request you to share them with us. Otherwise, we would greatly appreciate it if you could consider revising the score.
>
> We look forward to your response.
>
> Thank you,
>
> The Authors

---

> > ### Author Response · Authors · 2024-11-24
> > **Eagerly waiting for post-rebuttal feedback**
> >
> > Dear Reviewer D75u,
> >
> > Thanks again for your insightful and thoughtful comments!
> >
> > As the reviewer-author discussion period is closing soon (November 26 at 11:59 pm AoE), we would like to gently remind you that we are eagerly awaiting your feedback on our response.
> >
> > We have updated our revised manuscript, where we made the following updates as per your suggestions:
> >
> > - Assessing how well the generated materials align with the provided text prompts (Section 5.4).
> > - Comprehensive and detailed results for both Gen and CSP tasks across three benchmark datasets (Appendix F.3).
> >
> > If there are any remaining concerns, we kindly request you to share them with us. Otherwise, we would greatly appreciate it if you could consider revising the score.
> >
> > We look forward to your response.
> >
> > Thank you,
> >
> > The Authors

---

> > > ### Author Response · Authors · 2024-11-25
> > > **Eagerly waiting for your feedback**
> > >
> > > Dear Reviewer D75u,
> > >
> > > As the reviewer-author discussion period is closing soon (November 26 at 11:59 pm AoE), we would like to gently remind you that we are eagerly awaiting your feedback on our response and revised manuscript.
> > >
> > > We are happy to inform you that all of the three reviewers now lean towards acceptance. Your insights and evaluation play a crucial role in deciding the ultimate fate of our work, and we are eagerly awaiting your response to the revised manuscript.
> > >
> > > regards,
> > >
> > > Authors

---

> > > > ### Comment · Reviewer_D75u · 2024-11-25
> > > > **Response to the Authors**
> > > >
> > > > Thank you for the rebuttal
> > > >
> > > > I apprieciate the authors' efforts during rebuttal, especially the new comparison regarding the long and short textual prompt.
> > > >
> > > > I have raised the score to 6.

---

> > > > > ### Author Response · Authors · 2024-11-27
> > > > >
> > > > > Dear Reviewer D75u,
> > > > >
> > > > > We sincerely thank you for your positive feedback and for raising the score to 6. We have carefully incorporated all your suggested changes into our revised manuscript, and we believe these contributions significantly enhance the research at the intersection of AI and material science.
> > > > >
> > > > > We would be grateful if you could review the revised version and let us know if it is possible to change the decision from borderline accept to accept.
> > > > >
> > > > > Your feedback would be highly appreciated. If you feel that additional experiments or results would further strengthen the manuscript, we are more than willing to provide them and actively engage in any further discussions.
> > > > >
> > > > > Thank you once again for your valuable insights and support. We look forward to your response.
> > > > >
> > > > > Regards,
> > > > >
> > > > > The Authors

---

### Official Review · Reviewer_cwRr · 2024-11-08

**Soundness:** 4
**Presentation:** 3
**Contribution:** 3
**Rating:** 8
**Confidence:** 4

**Summary:**

This paper presents TGDMat, a novel text-guided diffusion model for generating periodic crystal materials. TDGMat jointly models the generation of atom types, coordinates, and lattices of materials using separate diffusion processes. Specifically, it uses a Denoising Diffusion Probabilistic Model for lattice modeling, and uses a discrete diffusion model D3PM for modeling atom types. As for atom coordinate modeling, the authors mainly follow DiffCSP and use a score matching objective. Note that, they apply diffusion on fractional coordinates, instead of Cartisian coordinates, which cannot reflect the periodicity of crystal materials. To ensure geometric symmetry, TGDMat uses the CSPNet proposed by DiffCSP, which ensures periodic E(3) invariance for periodic crystals.

For the text-guided component, material descriptions are generated using Robocrystallographer software. Text embeddings, produced by a language model, are then concatenated with node embeddings to guide material generation.

**Strengths:**

1. The overall performance is good, as shown in Table 3. TGDMat shows significant improvement on the random material generation task compared to previous methods. Additionally, the authors demonstrate the importance of incorporating text-guidance by showing the improved performance of baselines in Table 1 and Table 2.
2. The method is overall sound and well-engineered. The authors have employed state-of-the-art diffusion methods for each modality associated in the whole material generation process. The source code is attached as a supplementary material.
3. The textual annotation for the material dataset is a nice additional contribution, and is potentially impactful. It can stimulate future research for joint text and material modeling.

**Weaknesses:**

1. It seems that a large proportion of the methodology is borrowed from the previous work DiffCSP. This includes the diffusion process for atom coordinates, the diffusion for lattice, and the GNN backbone.
2. The authors are suggested to use the \citep command instead of \cite for citations to improve the readability.
3. The proposed method has achieved significant performance on the employed evaluation metrics, like validity and coverage. Does this mean the model can be readily employed for practical material discovery in industry? If yes, can you include further discussion on this application? If not, what is a barrier? Are there any other evaluation metrics that should be measured, like the novelty of the generated material compared to the training set, before application?

**Questions:**

1. How large is the proposed model and the compared baselines for material generation?

---

> ### Author Response · Authors · 2024-11-18
> **Point-by-point responses to Reviewer cwRr**
>
> Dear Reviewer cwRr,
>
> Thank you very much for providing us with valuable feedback. We appreciate the detailed comments. Below, we have provided point-by-point responses to each of your comments.
>
>
> Weaknesses:
> -
>
> ***Key differences between DiffCSP and TGDMat***
>
> > Following are the key differences between DiffCSP and our proposed TGDMat
> >
> >|           | DiffCSP          | TGDMat   |
> >| --------  | --------         | -------- |
> >| Tasks     | Only CSP Task    | Both CSP and Gen Tasks|
> >| Diffusion on Atom Type | NA    | Discrete Diffusion (D3PM)|
> >| Model Category | Unconditional; unable to specify the criteria required by the user.    | Conditional; able to specify the criteria required by the user. (in Text Format)|
> >| Text Guided Diffusion     | NO    | Yes|
> >
> >However, note the goal of this paper is not to introduce a new diffusion model to replace existing models like DiffCSP or CDVAE for periodic material generation. Instead, we focus on demonstrating that conditional models can outperform traditional unconditional models, such as DiffCSP. Specifically, we show that incorporating textual conditions through text-guided diffusion leads to better performance compared to using unconditional models like DiffCSP. Additionally, we enhance DiffCSP by integrating discrete diffusion over atom types in our proposed TGDMat framework.
>
>
> ***Suggestion:  Use of \citep command.***
>
> >Thanks for the suggestion. We will update the revised manuscript.
>
>
> ***Regarding Practical Deployment of the Model in Industry.***
>
> > Our proposed method demonstrates significant potential for practical material discovery in industry by generating valid, diverse, and structurally plausible materials aligned with user-provided textual descriptions. This capability positions our model as an efficient tool for creating “initial templates” of materials tailored for applications such as battery materials, solar cells, or catalysts, significantly reducing the time and computational resources required for exploratory studies.
> >
> >However, some barriers must be addressed before full deployment in industrial workflows. One challenge is the potential mismatch between generated structures and experimental ground truth, which arises from the inherent approximations in the model. These generated structures require further validation and refinement using computational methods, such as density functional theory (DFT), to ensure their physical and chemical feasibility before full deployment in industrial workflows.
>
>
> Questions:
> -
>
> ***Model Size of all Baselines and TGDMat***
>
>
> >| Models   | # Parameters    | Model size |
> >| -------- | --------        | --------   |
> >| CDVAE    | 4920414         | 18.771 MB  |
> >| SyMat    | 3385601         | 12.915 MB  |
> >| DiffCSP  | 12294656        | 46.923 MB  |
> >| TGDMat   | 12432228        | 47.448 MB  |
>
> We would like to once again express our gratitude to Reviewer cwRr for their valuable comments and suggestions. We will incorporate these insights into the revised manuscript. We believe our responses above effectively address all of Reviewer cwRr's concerns and further enhance the quality of our work.
>
> Sincerely,
> The Authors

---

> ### Author Response · Authors · 2024-11-23
> **Looking forward to your feedback**
>
> Dear Reviewer,
>
> Thank you for your valuable feedback and constructive comments.
>
> In our rebuttal, we have provided additional experiments and enhanced explanations, and we hope we have addressed all the concerns raised by the reviewer. We are open to further discussions and are happy to clarify any remaining doubts.
>
> If there are still any outstanding issues, we kindly request you to share them with us. Otherwise, we would greatly appreciate it if you could consider revising the score.
>
> We look forward to your response.
>
> Thank you,
>
> The Authors

---

> > ### Comment · Reviewer_cwRr · 2024-11-24
> > **Thank you for the response**
> >
> > Thank you for the response. I remain my original rating, as I did not see any updates on your submission.
> >
> > ICLR allows authors to revise their manuscript during rebuttal to resolve the reviewers' concerns. You can leverage this opportunity to improve your manuscript by incorporating the reviewers' comments. To help the reviewers' recognize your update, you can use use colored texts in your revised manuscript.

---

> > > ### Author Response · Authors · 2024-11-24
> > > **Response to Reviewer cwRr's Feedback**
> > >
> > > Dear Reviewer,
> > >
> > > Thank you for your valuable feedback and thoughtful comments.
> > >
> > > In response, we have uploaded the revised manuscript and made the following updates as per your suggestions:
> > >
> > > - Added a detailed discussion on the key differences between DiffCSP and TGDMat (Section 3 and Appendix B.5).
> > > - Replaced \cite commands with \citep for citations to improve the readability.
> > > - Included a model size comparison of baselines and TGDMat (Table 7, Appendix, Lines 1205–1212).
> > >
> > > We hope these revisions adequately address all the concerns you raised. If there are any remaining issues or additional clarifications needed, please let us know, and we would be happy to address them. Otherwise, we kindly request you to consider revising the score based on these updates.
> > >
> > > We look forward to your response.
> > >
> > > Thank you,
> > >
> > > The Authors

---

> > > > ### Author Response · Authors · 2024-11-25
> > > > **Keenly Seeking Your Feedback on Revised Manuscript**
> > > >
> > > > Dear Reviewer cwRr,
> > > >
> > > > As the reviewer-author discussion period is nearing its conclusion (November 26 at 11:59 pm AoE), we would like to kindly remind you that we are eagerly awaiting your feedback on our revised manuscript.
> > > >
> > > > We greatly value your insightful comments, which have been instrumental in improving our work. Your suggestions have been thoughtfully incorporated into the revision, and we hope the updated manuscript addresses all the concerns you raised.
> > > >
> > > > If there are any remaining issues or additional clarifications required, please do not hesitate to let us know—we would be happy to address them promptly. Otherwise, we kindly ask you to consider revising your score based on the updates.
> > > >
> > > > Thank you once again for your time and effort in reviewing our work. We look forward to hearing from you.
> > > >
> > > > Best regards,
> > > > The Authors

---

> > > > > ### Comment · Reviewer_cwRr · 2024-11-25
> > > > > **Followup feedback**
> > > > >
> > > > > Thank you for the revision. I think this paper would be a nice contribution to the ICLR conference. I have raised my scores accordingly.

---

### Author Response · Authors · 2024-11-24
**Summary of Revised Manuscript**

We sincerely thank the reviewers for their valuable insights and constructive feedback on our work. We have revised the main manuscript and appendix, with all changes highlighted in blue. A summary of the major updates is outlined as follows:

### Key Updates:
1. **Additional Experiments**:
   We have conducted all the additional experiments requested by the reviewers, including:
   - Assessing how well the generated materials align with the provided text prompts (Section 5.4).
   - Comprehensive and detailed results for both Gen and CSP tasks across three benchmark datasets (Appendix F.3).
   - Comparing the utility of text guidance versus feature vector guidance (Appendix F.7).
   - Ablation study on the joint learning of crystal geometry (Appendix F.8).

2. **Presentation Improvements**:
   We have implemented several enhancements to improve the clarity and readability of the manuscript, including:
   - Providing a detailed discussion on the key differences between DiffCSP and TGDMat.(Section 3 and Appendix B.5)
   - Replacing \cite commands with \citep for more consistent citation formatting.
   - Correcting a typo in Figure 2 regarding the direction of forward and reverse diffusions.

We believe these revisions comprehensively address the reviewers’ concerns and significantly improve the quality of our work.

---

### Meta-Review · Area_Chair_SDPu · 2024-12-23

**Metareview:**

This paper presents TGDMat, a text-guided diffusion model for generating periodic crystal materials. The proposed method integrates atom types, coordinates, and lattice parameters into a unified diffusion framework, guided by text-based descriptions. The reviewers appreciated the technical depth and relevance of the work, highlighting the innovative use of textual descriptions for guiding material generation. However, concerns were raised about the limited novelty compared to prior work, clarity in presenting the role of text guidance, and the robustness of the evaluation metrics. The authors addressed these issues through extensive rebuttals, additional experiments, and manuscript revisions, which convinced the reviewers of the method's contributions and applicability. With all reviewers leaning towards acceptance, the consensus supports the recommendation of accepting the paper as a poster presentation.

**Additional Comments On Reviewer Discussion:**

The discussion phase primarily focused on the method's novelty, the role of textual guidance compared to feature vector-based guidance, and the evaluation of generated materials' alignment with text prompts. The authors provided substantial clarifications, emphasizing the advantages of text embeddings in capturing complex material attributes, supported by additional experiments showing the superior performance of text-guided models. They also included ablation studies on joint versus separate learning of crystal parameters and a detailed comparison with related methods like DiFCSP. These updates addressed most reviewer concerns, leading to an improved consensus. Despite some residual doubts about broader applicability, the reviewers agreed that the work is a meaningful contribution to the field.

---

### Decision · Program_Chairs · 2025-01-22

Accept (Poster)